# GLOBAL OPTIMIZATION OF GRAPH ACQUISITION FUNCTIONS FOR NEURAL ARCHITECTURE SEARCH

## ABSTRACT

Graph Bayesian optimization (BO) has shown potential as a powerful and data-efficient tool for neural architecture search (NAS). Most existing graph BO works focus on developing graph surrogate models, i.e., metrics of networks and/or kernels to quantify the similarity between networks. However, optimization of the resulting acquisition functions over graph structures is less studied due to their complexity and formulations over the combinatorial graph search space. This paper presents explicit optimization formulations for graph input spaces, including properties such as reachability and shortest paths, which can then be used to formulate graph kernels and associated acquisition functions. We theoretically prove that the proposed encoding is an equivalent representation of the original graph space and provide a general formulation for neural architecture cells that incorporates node and/or edge-labeled graphs with multiple sources and sinks regardless of connectivity. Numerical results over several NAS benchmarks show that our method efficiently finds the optimal architecture for most cases.

## 1 INTRODUCTION

Despite numerous breakthroughs in deep learning, the design of neural architectures largely relies on prior experience and heuristic search. The field of neural architecture search (NAS) seeks systematic algorithms for designing the architecture of a neural network model (Ren et al., 2021). In general, NAS algorithms share several steps (Salmani Pour Avval et al., 2025): (i) encoding the search space, e.g., as a general or modular domain, (ii) prescribing a search strategy over the above space, and (iii) assessing the (approximate) performance at selected points. Early works in NAS sought to encode a general search space from scratch, e.g., as a string (Zoph & Le, 2017). Later works constrain the search space toward problem tractability, such as by explicitly encoding a layer- or module-based structure (Liu et al., 2018; Wu et al., 2019). Search strategies are often based on random search, gradient-based optimization (Liu et al., 2019; Wu et al., 2019), Bayesian optimization (Ru et al., 2021; White et al., 2021a), evolutionary algorithms (Real et al., 2019; Qiu et al., 2023), or reinforcement learning (Zoph & Le, 2017; Jaafra et al., 2019; Cheng et al., 2022). Finally, performance assessments are the most expensive step of NAS, often involving full or partial training of the proposed model(s).

Graph Bayesian optimization (BO) exhibits particular promise for NAS (Elsken et al., 2019; White et al., 2023), given its efficiency in exploring the graph search space, i.e., treating the model as a directed graph, and identifying promising architectures within limited budgets (Ru et al., 2021). Graph BO addresses the above NAS steps using (i) a trained graph surrogate that serves as a predictor, and (ii) an acquisition function, encoding trade-offs between exploitation and exploration, which is then optimized to give the next candidate to sample. From modeling perspectives, Gaussian processes (GPs) (Schulz et al., 2018) are commonly used, since they offer accurate prediction along with principled uncertainty quantification. To apply GPs in a graph domain, graph kernels (Vishwanathan et al., 2010; Borgwardt et al., 2020; Kriege et al., 2020; Nikolentzos et al., 2021) are introduced to measure the similarity between graphs. Although advances in graph kernels facilitate the generalization from non-structural spaces to graph space, optimizing the resulting acquisition functions over (combinatorial) graph spaces remains a challenge. Most works use sample-based or evolutionary algorithms due to cheap evaluations of the acquisition function, but must incorporate problem-specific constraints into the sampling and mutation steps to remove invalid candidates, and lack theoretical guarantees about the optimality of solutions.

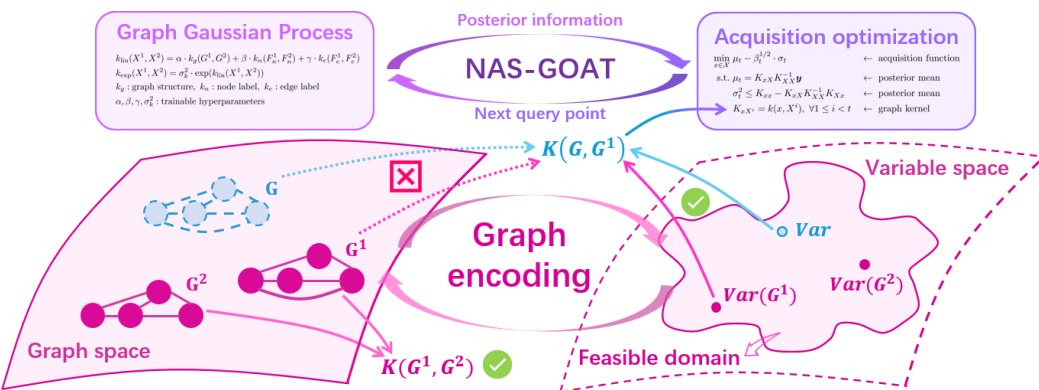

Figure 1: Illustration of NAS-GOAT. The main idea is to represent graphs in variable space and introduce constraints to build a bijection between all graphs and the feasible domain. The graph kernel value between an unknown graph (which is our optimization target) and a given graph is then formulated as expressions of variables, or constraints, enabling us to employ global optimization for acquisition function and propose the next neural architecture to evaluate.

Recently, the idea of using mathematical programming techniques to formulate machine learning (ML) models, e.g., neural networks (NNs) (Fischetti & Jo, 2018; Anderson et al., 2020; Tsay et al., 2021; Zhang et al., 2023), trees (Mišić, 2020; Mistry et al., 2021; Ammari et al., 2023), and GPs (Schweidtmann et al., 2021; Xie et al., 2024), provides a way to explicitly solve decision-making problems involving ML models. Relevant applications include BO acquisition optimization (Thebelt et al., 2021; 2022; Wang et al., 2023), NN verification (Hojny et al., 2024), and molecular design (McDonald et al., 2024; Zhang et al., 2024), among others. Based on the global optimization formulation for acquisition optimization proposed in Xie et al. (2024), Xie et al. (2025) propose BoGrape as a general graph BO framework, comprising the first work to treat graph acquisition functions from a discrete optimization viewpoint. By encoding graph spaces and shortest-path graph kernels (Borgwardt & Kriegel, 2005) into mixed-integer programming (MIP), BoGrape can handle constraints over graph search spaces and globally optimize the acquisition function with mathematical guarantees. However, the requirement of strong connectivity makes BoGrape unsuitable for NAS, since neural architectures usually include weakly connected acyclic digraphs (DAGs).

This paper studies the global optimization of graph acquisition functions for graph BO-based NAS. To represent the graph space containing valid neural architectures, we theoretically generalize the graph encoding presented in Xie et al. (2025) to omit assumptions about connectivity, and further restrict the general encoding to the NAS search space. GPs with shortest-path kernels are used as graph surrogates, and lower confidence bound (LCB) (Srinivas et al., 2010) as the acquisition function. The proposed graph encoding contains more graph properties than Xie et al. (2025), including reachability, shortest distances and shortest paths, and is compatible with existing formulations for shortest-path graph kernels and acquisition functions. The final acquisition optimization is formulated as a MIP, which can be solved with global optimality guarantees. Figure 1 illustrates the main idea of the proposed framework; we also list the major contributions of this work here:

- We present an equivalent representation for general labeled graphs in optimization variable space. Each graph corresponds to a unique feasible solution containing its graph structure, as well as graph properties like reachability, shortest distances, and shortest paths.

- We provide a general kernel form measuring the similarity between two labeled graphs over graph structure, node label, and edge label levels, and we present a formulation that is compatible with our graph encoding.

- We incorporate NAS-specific constraints to the graph encoding, which handles settings with multiple sinks/sources, edge and node labels regardless of connectivity in NAS tasks.

- We propose NAS-GOAT to **g**lobally **o**ptimize graph **a**cquisition functions based on our proposed encoding. Numerical results demonstrating a full BO loop on NAS benchmarks show the efficiency and potential of NAS-GOAT.

## 2 BACKGROUND

### 2.1 CELL-BASED NAS

In many NAS search spaces, a network architecture is designed by varying some repeated small feedforward sub-structures termed *cells* (Ying et al., 2019; Dong & Yang, 2020). Each cell is treated as a DAG, where the operation units are represented as node or edge labels, and information flows within the cell following graph topologies. Cells are then stacked multiple times and embedded into a macro neural network skeleton to give the final architecture. For instance, NAS-Bench-101 (Ying et al., 2019) and NAS-Bench-201 (Dong & Yang, 2020) define one stack as 3 and 5 replications of cells, respectively, and each stack appears 3 times in the overall network structure. In more challenging cases such as NAS-Bench-301 (Zela et al., 2022), cells may not be identical. Cell-based NAS can be effectively considered as an expensive black-box optimization problem over a graph input domain, where one seeks the best graph, i.e., cell, that optimizes the performance of the resulting neural architecture over certain metrics, e.g., validation/test accuracy.

### 2.2 GRAPH BAYESIAN OPTIMIZATION

Graph BO is a natural extension of BO (Frazier, 2018; Garnett, 2023) from vector space to graph space. At the $t$-th iteration, a graph Gaussian process (GP) equipped with a graph kernel is trained on available data $X = \{(G^i, F^i), y^i\}_{i=1}^{t-1}$. The posterior distribution of the graph GP is then used to define acquisition functions such as lower confidence bound (LCB): $\alpha_{LCB}(x) = \mu_t(x) - \beta_t^{1/2} \cdot \sigma_t(x)$, where $\beta_t$ is a hyperparameter balancing between exploitation and exploration.

From modeling perspectives, the core component of graph GPs is the graph kernel that measures similarity between graphs. Classic graph kernels include random walk (RW) (Gärtner et al., 2003), subgraph matching (SM) (Kriege & Mutzel, 2012), shortest-path (SP) (Borgwardt & Kriegel, 2005), Weisfeiler-Lehman (WL) (Shervashidze et al., 2011), and Weisfeiler-Lehman optimal transport (WLOA) (Kriege et al., 2016) kernels. We refer the reader to Vishwanathan et al. (2010); Borgwardt et al. (2020); Kriege et al. (2020); Nikolentzos et al. (2021) for comprehensive details about graph kernels. In this work, we consider SP kernels used for graph BO in Xie et al. (2025). Mathematically, for two node-labeled graphs $G^1$ and $G^2$, denote $V^1$ and $V^2$ as their node sets, respectively, $l_v$ as the label of $v$, and $d_{u,v}$ as the shortest distance from node $u$ to node $v$. The SP kernel is defined as:

$$k_{SP}(G^1, G^2) = \frac{1}{n_1^2 n_2^2} \sum_{u_1, v_1 \in V^1, u_2, v_2 \in V^2} \mathbf{1}(l_{u_1} = l_{u_2}) \cdot \mathbf{1}(d_{u_1,v_1} = d_{u_2,v_2}) \cdot \mathbf{1}(l_{v_1} = l_{v_2}), \quad (k_g)$$

where $n_1^2 n_2^2$ is a normalizing coefficient with $n_1$ and $n_2$ as the node number of $G^1$ and $G^2$, resp.

### 2.3 GRAPH ACQUISITION OPTIMIZATION

The major challenge of graph BO is the acquisition optimization step, which seeks to find the graph structure (i.e., connectivity, nodes, labels) with optimal acquisition function value and is often required for most BO convergence proofs. Encoding a graph search space and acquisition function as optimization constraints is non-trivial, and most existing works follow a sample-then-evaluate procedure to avoid directly optimizing over discrete space, e.g., see Kandasamy et al. (2018); Ru et al. (2021); Wan et al. (2021; 2023). From a discrete optimization viewpoint, Xie et al. (2025) first formulate the space of strongly connected graphs using MIP, and propose BoGrape as a graph BO framework that can globally optimize the lower confidence bound (LCB) acquisition:

$$
\begin{aligned}
&\min_{x \in \mathcal{X}} \mu_t - \beta_t^{1/2} \cdot \sigma_t && \leftarrow \text{ acquisition function} \\
&\text{s.t. } \mu_t = K_{xX} K_{XX}^{-1} \boldsymbol{y} && \leftarrow \text{ posterior mean} \\
&\quad\ \sigma_t^2 \leq K_{xx} - K_{xX} K_{XX}^{-1} K_{Xx} && \leftarrow \text{ posterior mean} \\
&\quad\ K_{xX^i} = k(x, X^i), \ \forall 1 \leq i < t && \leftarrow \text{ graph kernel}
\end{aligned}
\quad \text{(Acq-Opt)}
$$

However, NAS settings involve graphs that are weakly connected, or even disconnected, potentially with edge labels. Embedding this general graph domain in MIP remains an open challenge.

Table 1: Variables introduced to encode shortest paths for an arbitrary graph. Since the shortest distance between two nodes is strictly less than $n$, we use $d_{u,v} = n$ to denote that node $u$ cannot reach node $v$.

| Variables | Domain | Description |
|---|---|---|
| $A_{v,v}, \ v \in [n]$ | $\{0, 1\}$ | if node $v$ exists |
| $A_{u,v}, \ u, v \in [n], \ u \neq v$ | $\{0, 1\}$ | if edge $u \rightarrow v$ exists |
| $r_{u,v}, \ u, v \in [n]$ | $\{0, 1\}$ | if node $u$ can reach node $v$ |
| $d_{u,v}, \ u, v \in [n]$ | $[n + 1]$ | the shortest distance from node $u$ to node $v$ |
| $\delta_{u,v}^{w}, \ u, v, w \in [n]$ | $\{0, 1\}$ | if node $w$ appears on the shortest path from node $u$ to node $v$ |

## 3 METHODOLOGY

### 3.1 ENCODE A NAS GRAPH SEARCH SPACE IN OPTIMIZATION

This paper precisely seeks a MIP encoding for the general search space over graphs found in NAS settings. In other words, we must formally define the graph search space over which acquisition optimization is performed. For the sake of exposition, we temporarily ignore node/edge features and first focus on an optimization formulation over variable graph structures. To avoid graph isomorphism caused by node indexing, we assume that all nodes are labeled differently. We discuss node/edge features in Section 3.3. Intuitively, encoding such a general graph space is easy, since each graph is uniquely determined by its adjacency matrix, and one simply needs to define the $n \times n$ adjacency matrix containing binary variables $A_{u,v}$ that denote the existence of edge $u \rightarrow v$. However, this naive encoding omits important graph information, e.g., connectivity, reachability, and shortest distance between nodes, which are important for defining acquisition functions and constraining feasible graphs. Encoding these graph properties into the space of decision variables is significantly more challenging because we must define constraints that mathematically prescribe all variables to take correct values for *any* possible graph in the search space.

The graph encoding introduced in this paper incorporates reachability, shortest distances, and shortest paths for any graph without requiring strong connectivity (or in fact any connectivity requirements) as in previous work (Xie et al., 2025). We describe the mathematical differences in detail between these formulations in Appendix A.3. These metrics are then used to encode shortest-path graph kernels for graph BO. To begin with, we define variables corresponding to relevant graph properties in Table 1. We consider all graphs with node number ranging from $n_0$ to $n$. For simplicity, we use $[n]$ to denote the set $\{0, 1, \ldots, n - 1\}$.

For each variable $Var$ in Table 1 , we use $Var(G)$ to denote its value on a given graph $G$. For example, $d_{u,v}(G)$ is the shortest distance from node $u$ to node $v$ in graph $G$. If graph $G$ is given, all variable values can be easily obtained using classic shortest-path algorithms, such as the Floyd–Warshall algorithm (Floyd, 1962). However, for optimization over arbitrary graphs, the variables must be properly defined using mathematical constraints, such that they take correct values for any given graph, i.e., to match the `Description` column in Table 1. For conciseness, we only present our final derived encoding in Eq. (Graph-Encoding) and the major theoretical result in Theorem 1. Full derivations of our formulation and its correctness are given in Appendix A.

Eq. (Graph-Encoding) comprises many linear constraints encoding correctness of shortest paths for any graphs in the search space, specifically defined as satisfying Conditions $(\mathcal{C}1)$–$(\mathcal{C}8)$ in Appendix A.1. Here we present the final formulation, which conveys the overall idea about how to use constraints to mathematically define variables over graphs. Constraints for optimization formulations must be carefully selected. There are often multiple ways to encode a combinatorial problem, but insufficient constraints result in an unnecessarily large search space with symmetric solutions, while excessive constraints may cutoff feasible solutions from the search space. Theorem 1 guarantees that our encoding exactly formulates the graph space (see Appendix A.2 for proofs).

**Theorem 1.** *There is a bijection between the feasible domain restricted by Eq. (Graph-Encoding) with size $[n_0, n]$ and the complete graph space with node numbers in $[n_0, n]$.*

$$
\begin{cases}
\displaystyle\sum_{v \in [n]} A_{v,v} \geq n_0 \\[2mm]
A_{v,v} \geq A_{v+1,v+1} \\
2 \cdot A_{u,v} \leq A_{u,u} + A_{v,v} \\
2 \cdot r_{u,v} \leq A_{u,u} + A_{v,v} \\
d_{u,v} \geq n \cdot (1 - A_{u,u}) \\
d_{u,v} \geq n \cdot (1 - A_{v,v}) \\
r_{v,v} = 1 \\
d_{v,v} = 0 \\
\delta_{v,v}^{v} = 1 \\
\delta_{v,v}^{w} = 0 \\
r_{u,v} \geq A_{u,v} \\
d_{u,v} \geq 2 - A_{u,v} \\
d_{u,v} \leq 1 + (n-1) \cdot (1 - A_{u,v}) \\
d_{u,v} \leq n - r_{u,v} \\
d_{u,v} \geq n - (n-1) \cdot r_{u,v} \\
r_{u,w} + r_{w,v} \geq 2 \cdot \delta_{u,v}^{w} \\
r_{u,v} \geq r_{u,w} + r_{w,v} - 1 \\
\delta_{u,v}^{u} = \delta_{u,v}^{v} = 1 \\
\displaystyle\sum_{w \in [n]} \delta_{u,v}^{w} \geq 2 + r_{u,v} - A_{u,v} \\[2mm]
\displaystyle\sum_{w \in [n]} \delta_{u,v}^{w} \leq 2 + (n-2) \cdot (r_{u,v} - A_{u,v}) \\[2mm]
d_{u,v} \leq d_{u,w} + d_{w,v} - (1 - \delta_{u,v}^{w}) + (n+1) \cdot (2 - r_{u,w} - r_{w,v}) \\
d_{u,v} \geq d_{u,w} + d_{w,v} - 2n \cdot (1 - \delta_{u,v}^{w})
\end{cases}
\qquad \text{(Graph-Encoding)}
$$

### 3.2 FROM GRAPH TO CELL IN NEURAL ARCHITECTURE SEARCH

Eq. (Graph-Encoding) defines general graph topology, while cells considered in NAS are DAGs. In practice, these cells have more specific graph structures and features, requiring additional constraints to further restrict the feasible domain. Based on Eq. (Graph-Encoding), we present these constraints in Eq. (1) to formulate cells that are node- and/or edge-labeled DAGs with multiple sources (input nodes) and sinks (output nodes). Note that cells could be disconnected.

Consider graphs with $n$ nodes indexed by $[n]$, among which $I \subset [n]$ and $O \subset [n]$ are source and sink indices, respectively. Let $L_n$ denote the number of node labels (including two extra labels to identify the sources and the sinks) and $L_e$ the number of edge labels. W.l.o.g., in terms of node labels, we use the first label for the sources, and the last label for the sinks. Introducing variable $F_{v,l} \in \{0, 1\}$ to represent whether node $v \in [n]$ has label $l \in [L_n]$, and variable $F_{u \to v,l}$ to represent whether edge $u \to v$ (with $u < v$) has label $l \in [L_e]$, we recover the encoding in Eq. (1).

Eq. (1a) enforces that each edge starts from the node with smaller index to reduce the number of isomorphic graphs. Eqs. (1b)–(1c) are definitions of sources (zero in-degree), and sinks (zero out-degree), resp. Eqs. (1d) and (1f) set nodes with indices in $I$ as the sources, where any other nodes can be reached by at least one source. Similarly, Eqs. (1e) and (1g) define nodes indexed by $O$ as the sinks, where any other nodes can reach to at least one sink. When the cell has a single source or a single sink, Eqs. (1d) and (1e) implicitly enforce the weak connectivity of the cell, which is the most classic setting in cell-based NAS. Eq. (1h) enforces each node to take one label. Eq. (1i) forces one edge label for each existing edge and no edge labels for nonexistent edges.

Adding all the constraints in Eq. (1) to Eq. (Graph-Encoding) produces a feasible domain containing all the node- and edge-labeled DAGs with multiple sources and sinks (not necessarily connected). In practice, the encoding can easily become more general by simply removing unnecessary constraints. For example, cells defined in NAS-Bench-201 datasets are weakly connected edge-labeled DAGs, meaning Eqs. (1f)–(1h) are unnecessary. Other benchmark-specific restrictions on cells can also be seamlessly added to the encoding, such as limits on the number of edges, e.g., NAS-Bench-101, and disconnected graphs formulated as two cells, e.g., NAS-Bench-301. Appendix C details how these two cases are handled.

$$
\begin{cases}
d_{u,v} = n, \ \forall u, v \in [n], \ u > v & \text{(1a)} \\
d_{v,i} = n, \ \forall i \in I, \ v \in [n] \backslash I & \text{(1b)} \\
d_{o,v} = n, \ \forall o \in O, \ v \in [n] \backslash O & \text{(1c)} \\
\sum_{i \in I} r_{i,v} \geq 1, \ \forall v \in [n] \backslash I & \text{(1d)} \\
\sum_{o \in O} r_{v,o} \geq 1, \ \forall v \in [n] \backslash O & \text{(1e)} \\
F_{i,0} = 1, \ F_{v,0} = 0, \ \forall i \in I, \ v \in [n] \backslash I & \text{(1f)} \\
F_{o,L_{n-1}} = 1, \ F_{v,L_{n-1}} = 0, \ \forall o \in O, \ v \in [n] \backslash O & \text{(1g)} \\
\sum_{l \in [L_n]} F_{v,l} = 1, \ \forall v \in [n] & \text{(1h)} \\
\sum_{l \in [L_e]} F_{u \to v,l} = A_{u,v}, \ \forall u, v \in [n], \ u \neq v & \text{(1i)}
\end{cases}
$$

### 3.3 Encode Graph Kernels

We take the triple $(G, F_n, F_e)$ as a graph with node labels $F_n = \{F_{v,l}\}_{v \in [n], \ l \in [L_n]}$ and edge labels $F_e = \{F_{u \to v,l}\}_{u,v \in [n], \ l \in [L_e]}$. Given two graphs $X^1 = (G^1, F_n^1, F_e^1)$ and $X^2 = (G^2, F_n^2, F_e^2)$ with node numbers $n_1$ and $n_2$, resp., we define the following general kernel form:

$$
k_{\lin}(X^1, X^2) = \alpha \cdot k_g(G^1, G^2) + \beta \cdot k_n(F_n^1, F_n^2) + \gamma \cdot k_e(F_e^1, F_e^2), \qquad \text{(linear)}
$$

where kernels $k_g, k_n, k_e$ quantify similarity over graph structure, node labels, and edge labels, resp.

We then denote the optimization target (i.e., to maximize the acquisition function) as an unknown graph $x = (G, F_n, F_e)$, and the available data points as $X = \{X^i, y^i\}_{i=1}^{t-1}$ with $X^i = (G^i, F_n^i, F_e^i)$. After properly defining the search space in Section 3.2, we must encode kernel-relevant terms in Eq. (Acq-Opt), i.e., $k_{xX^i}$ and $k_{xx}$. We take a similar approach to Xie et al. (2025) to encode the graph structure kernel, i.e., $k_g(G, G), k_g(G, G^i)$ and binary node features, i.e., $k_n(F_n, F_n), k_n(F_n, F_n^i)$ for graph structure $G$ and node labels $F_n$. Formulations are given in Appendix B.

**Edge label encoding:** Edge labels can be treated in a similar way to node labels. However, several NAS settings have more specific properties, i.e., all nodes are indexed when edge labels are present, and all graphs have the same size. Therefore, we propose the following alternative form:

$$
k_e(F_e^1, F_e^2) = \frac{2}{n(n-1)} \langle F_e^1, F_e^2 \rangle = \frac{2}{n(n-1)} \sum_{u < v} \sum_{l \in [L_e]} F_{u \to v,l}^1 \cdot F_{u \to v,l}^2, \qquad (k_e)
$$

where $n(n-1)/2$ is a normalizing coefficient, with $n$ as the node number of both $G^1$ and $G^2$, given that a DAG has at most $n(n-1)/2$ edges.

We take edge kernels as follows, and evaluate their performance in Section 4.3:

$$
k_e(F_e, F_e^i) = \frac{2}{n(n-1)} \sum_{u < v} \sum_{l \in [L_e]} F_{u \to v,l}^i \cdot F_{u \to v,l},
$$

$$
k_e(F_e, F_e) = \frac{2}{n(n-1)} \sum_{u < v} \sum_{l \in [L_e]} F_{u \to v,l}^2 = \frac{2}{n(n-1)} \sum_{u < v} \sum_{l \in [L_e]} F_{u \to v,l} = \frac{2}{n(n-1)} \sum_{u < v} A_{u,v},
$$

where we use the trick that $x^2 = x$ for binary $x$ and the relation in Eq. (1i).

The above defines a formulation for all relevant terms in kernel form (linear). To improve representation ability, we also consider an alternative exponential form defined as:

$$
k_{\exp}(X^1, X^2) = \sigma_k^2 \cdot \exp(k_{\lin}(X^1, X^2)), \qquad \text{(exponential)}
$$

where the variance $\sigma_k^2$ controls the magnitude of kernel values.

## 4 EXPERIMENTS

All experiments are performed on a 4.7 GHz Intel Core i7-1260P CPU with 32 GB memory. For our methods, we use GPflow (Matthews et al., 2017) to implement GP models, and Gurobi (Gurobi Optimization, LLC, 2024) to solve MIPs. For kernel comparison, GraKel (Siglidis et al., 2020) is used to implement graph kernels. We use the published implementations of NAS-BOWL (Ru et al., 2021) and Naszilla (White et al., 2020; 2021b;a) for all other NAS baselines.

### 4.1 BENCHMARKS

We evaluate the performance of our graph BO-based method using the most popular benchmarks used in NAS literature: NAS-Bench-101 (Ying et al., 2019), NAS-Bench-201 (Dong & Yang, 2020), and NAS-Bench-301 (Zela et al., 2022). The former two benchmarks correspond to the classic node- and edge-labeled DAGs cases, respectively, where the cells involved in an architecture are identical. NAS-Bench-301 represents a more challenging setting, with larger graph sizes and more edge labels. Moreover, the overall architecture involves two cell designs, resulting in a disconnected edge-labeled graph search space during optimization. Details about these benchmarks are provided in Appendix C.

**NAS-Bench-101 (N101):** DAGs with one source, one sink, at most 7 nodes and 9 edges, and 3 different node operations. Only the source is labeled as operation IN, only the sink is labeled as operation OUT, and each of other nodes has one of the remaining three operations: 3x3 convolution, 1x1 convolution, or 3x3 max pooling. After removal of duplicates, N101 has approximately 423k unique architectures. Each architecture is trained on CIFAR-10 to obtain validation and test accuracies.

**NAS-Bench-201 (N201):** Dense DAGs with 4 nodes. Each of the 6 edges has a label chosen from 5 operation types: zeroize, skip-connection, 1x1 convolution, 3x3 convolution, or 3x3 average pooling. N201 has 15,625 architectures in total, each of which has various metrics including validation and test accuracies over three datasets: CIFAR10, CIFAR100, and ImageNet-16-120.

**NAS-Bench-301 (N301):** A surrogate NAS benchmark on the DARTS search space (Liu et al., 2019). The normal cell and reduction cell in DARTS architecture each defines a DAG with 7 nodes (2 sources, 4 intermediate nodes and 1 sink) and 12 edges. Each edge between the source and an intermediate node is labeled with one of the 8 operations: zeroize, identity, skip-connection, $3 \times 3$ and $5 \times 5$ separable convolutions, $3 \times 3$ and $5 \times 5$ dilated separable convolutions, $3 \times 3$ max pooling, $3 \times 3$ average pooling, identity, and zero. The two cells are treated as one disconnected graph for optimization since they may not be identical. N301 predicts the validation accuracies of different architectures on CIFAR-10 dataset.

For N101 and N201 benchmarks, each architecture is trained 20 times with varying random seeds, which could be used as a noisy objective function as suggested in Ru et al. (2021). For N301, the noise comes from prediction variance of different ensembles. We conduct experiments on the three benchmarks, reporting results for both the deterministic setting, i.e., averaging the accuracies over multiple random seeds/ensembles, and the noisy setting.

Table 2: GP model performance metrics using different graph kernels. For each dataset, 50 and 400 architectures are sampled for training and testing, resp. Predictive performance metrics are averaged over 20 replications and reported in the table, with one standard deviation in the brackets. The best method is marked in **bold** metric-wisely for each dataset.

| Graph type | Node-labeled DAG (N101) | | | Edge-labeled DAG (N201) | | |
|---|---|---|---|---|---|---|
| Kernel | RMSE ↓ | MNLL ↓ | Spearman ↑ | RMSE ↓ | MNLL ↓ | Spearman ↑ |
| RW | 0.29(0.01) | 30.29(5.26) | 0.81(0.04) | 0.32(0.02) | 43.07(19.14) | 0.78(0.05) |
| WL | 0.15(0.02) | **-0.77(0.07)** | 0.87(0.03) | **0.23(0.04)** | 0.98(1.02) | **0.81(0.07)** |
| WL-edge | - | - | - | 0.37(0.02) | 46.00(16.73) | 0.11(0.09) |
| SP | 0.21(0.05) | 227.67(114.59) | 0.83(0.04) | 0.33(0.04) | 465.15(152.20) | 0.63(0.10) |
| ESP | **0.11(0.02)** | 28.83(15.02) | **0.93(0.02)** | 0.30(0.04) | **0.36(0.22)** | 0.64(0.11) |

## 4.2 BASELINES

We compare our method, NAS-GOAT, which is capable of **g**lobally **o**ptimizing **a**cquisi**t**ion in form (Acq-Opt) with the encoding introduced in Section 3, against state-of-the-art baselines in NAS, described in Table 3 of Appendix D.2. BO-based baselines either use GPs or neural predictors as the surrogate model. Graph inputs are featurized into vectors using different encoding methods before being input to NN surrogates (Snoek et al., 2015; Springenberg et al., 2016; Shi et al., 2020; White et al., 2021a). For GP surrogate models, graphs can be directly used as data points by defining a proper graph kernel (Ru et al., 2021) or graph similarity metric (Kandasamy et al., 2018). In addition to BO-based algorithms, we also include popular methods in NAS such as random search, regularized evolution (Real et al., 2019), local search (White et al., 2021b) and GCN predictor (Wen et al., 2020). For BO-based methods, optimization of acquisition functions is achieved through mutation or sampling, while NAS-GOAT is capable of globally acquisition optimization over graph search spaces using MIP. Complete descriptions and implementation details of the baselines can be found in Appendix D, and a comparison against recent non-BO-based NAS methods is given in Appendix D.4.

## 4.3 GRAPH KERNELS COMPARISON

Although the focus of our work is a MIP formulation for global acquisition optimization, the GP model performance remains important to overall BO performance, noting that we employ SP kernels. In this section, we compare the predictive performance of graph GPs equipped with various graph kernels. For DAGs with node labels (N101), we compare RW, WL, and our kernels in form (linear) (SP) and (exponential) (ESP). For DAGs with edge labels (N201), all architectures are first converted to node-labeled graphs and then evaluated using RW and WL kernels. We also test the performance of WL kernels over the original edge-labeled graphs (denoted as WL-e). Both SP and ESP kernels can directly handle edge labels without conversion.

Table 2 reports performance metrics including root mean squared error (RMSE) and Spearman's rank correlation (Spearman) showing the predictive accuracy, as well as mean negative log likelihood (MNLL). Appendix D provides visualizations of GP predictions with different kernels on both node- and edge-labeled DAGs sampled from N101 and N201, respectively. The RW kernel does not perform well on either case. The WL kernel performs significantly better on converted node-labeled graphs compared to the original edge-labeled graphs, which matches the empirical observations in Ru et al. (2021). WL and ESP kernels exhibit the best overall performance, notably better than the simpler SP kernel. We found that a better kernel does not necessarily translate to better BO performance, since a simpler kernel may have benefits during the acquisition optimization step, resulting in a computational trade-off between the modeling and optimization steps.

## 4.4 GRAPH BO FOR NAS

Following the batch setting in Ru et al. (2021); White et al. (2021a), we conduct 30 BO iterations starting with 10 initial samples. At each iteration, we solve the MIP defined by Eq. (Graph-Encoding) and Eq. (1) using Gurobi (Gurobi Optimization, LLC, 2024) and store the best 5 candidates (in terms of acquisition function value) to evaluate.

We denote our methods as NAS-GOAT-L and NAS-GOAT-E to differentiate using the (linear) and (exponential) kernels, respectively, in graph GP. All baselines introduced in Table 3 are implemented, but here we only report Random, GCN (Wen et al., 2020), Evolution (Real et al., 2019), NAS-BOT (Kandasamy et al., 2018), BANANAS (White et al., 2021a) and NAS-BOWL (Ru et al., 2021) for clarity, since they generally achieve better results. Full baseline results are given in Appendix D.3. Not all the baselines are available for N301, we only report the available ones here. Specifically, N301 involves designing two separate cells, which NAS-GOAT can handle simultaneously as a disconnected graph. Other methods cannot directly handle this setting, e.g., the competing methods BANANAS and Evolution iterate between improving the two cells.

Following NAS literature, we minimize over validation error and report both validation and test errors (except for N301 where only validation errors are available). In other words, in terms of the BO algorithm, performance is only evaluated using the validation error as the black-box function to be optimized. We show the results in terms of validation error in Figures 2–3 and test error in Figure 8 of Appendix D.3. Differences between validation- and test-error performance may be

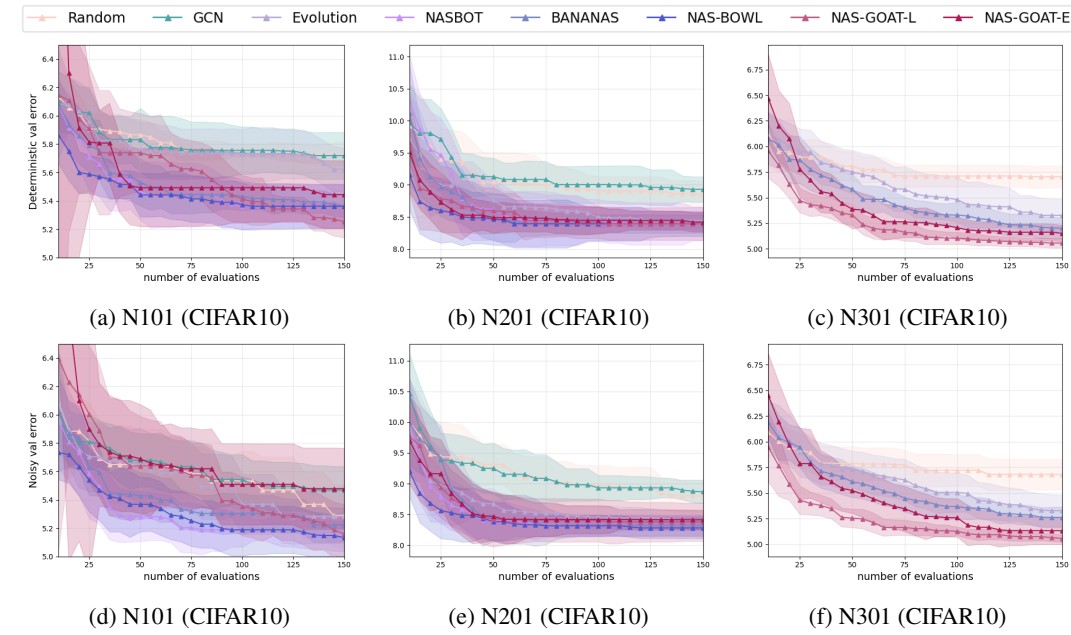

Figure 2: Numerical results on N101, N201 and N301 for CIFAR10. **Top:** Deterministic validation error. **Bottom:** Noisy validation error. Median with one std. deviation over 20 replications is plotted.

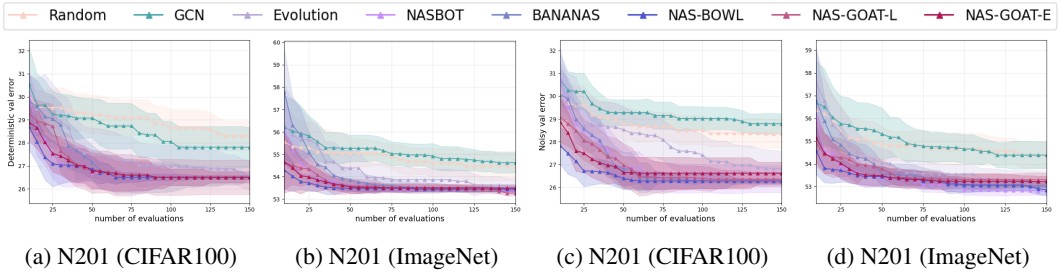

Figure 3: Numerical results on N201 for other datasets. **(a)-(b):** Deterministic validation error. **(c)-(d):** Noisy validation error. Median with one standard deviation over 20 replications is plotted.

improved by larger validation datasets. In general, both NAS-GOAT-L and NAS-GOAT-E find (near-)optimal architectures in terms of validation error. NAS-GOAT-L achieves slightly better performance, perhaps owing to its simpler form and resulting optimization formulation. Figure 2 evaluates all three benchmarks on the CIFAR10 dataset. Notably, NAS-GOAT considerably outperforms all baselines in the most challenging N301 setting, highlighting its importance as a general NAS framework that can be adapted to more challenging settings. Specifically, N301 reveals the potential of NAS-GOAT past the simpler N101 and N201 benchmarks, which are inherently limited by considering only the design of a single repeated cell. Nevertheless, on these benchmarks NAS-GOAT achieves similar efficiency to other BO methods (Figures 2–3) and final performance to other NAS methods (Appendix D).

## 5 CONCLUSIONS

This work considers global acquisition optimization in graph BO for NAS. The graph search space is precisely encoded into an equivalent variable space for discrete optimization. A general kernel is designed to handle both node and edge labels, and formulations are proposed based on the graph encoding. After adding suitable constraints to remove invalid architectures, we are able to globally optimize the acquisition function at each BO iteration, demonstrating promising results on commonly used NAS benchmarks. Future works could consider more graph kernels beyond shortest-path kernels, or apply the proposed method to more graph-based decision making problems.

REPRODUCIBILITY STATEMENT

We ensure our results are reproducible by providing theoretical proofs, code implementations, and documentation. The complete formulations are presented in Section 3 and Appendices A–C. Proofs to all the theorems in this paper can be found in the Appendix A.2. We provide code implementation of our method, NAS-GOAT, along with instructions for replicating the experiments in Section 4 in the supplementary materials.

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

# A SHORTEST PATH ENCODING

## A.1 ENCODING

For each variable $Var$ in Table 1 , we use $Var(G)$ to denote its value on a given graph $G$. For example, $d_{u,v}(G)$ is the shortest distance from node $u$ to node $v$ in graph $G$. If graph $G$ is given, all variable values can be easily obtained using classic shortest-path algorithms like the Floyd–Warshall algorithm (Floyd, 1962). However, for graph optimization, those variables need to be constrained properly so that they have correct values for any given graph. In this section, we first provide a list of necessary conditions that these variables should satisfy based on their definitions. Then we will prove that these conditions are sufficient in next section.

**Condition ($\mathcal{C}1$):** At least $n_0$ nodes exist. W.l.o.g., assume that nodes with smaller indexes exist:

$$\begin{cases} \sum_{v \in [n]} A_{v,v} \geq n_0 \\ A_{v,v} \geq A_{v+1,v+1}, \quad \forall v \in [n-1] \end{cases}$$

**Condition ($\mathcal{C}2$):** Initialization for nonexistent nodes, i.e., if either node $u$ or node $v$ does not exist, edge $u \to v$ cannot exists, node $u$ cannot reach node $v$, and the shortest distance from node $u$ to node $v$ is infinity, i.e., $n$:

$$\min(A_{u,u}, A_{v,v}) = 0 \Rightarrow A_{u,v} = 0,\ r_{u,v} = 0,\ d_{u,v} = n,\ \forall u,v \in [n],\ u \neq v$$

which could be rewritten as the following linear constraints:

$$\begin{cases} 2 \cdot A_{u,v} \leq A_{u,u} + A_{v,v}, & \forall u,v \in [n],\ u \neq v \\ 2 \cdot r_{u,v} \leq A_{u,u} + A_{v,v}, & \forall u,v \in [n],\ u \neq v \\ d_{u,v} \geq n \cdot (1 - A_{u,u}), & \forall u,v \in [n],\ u \neq v \\ d_{u,v} \geq n \cdot (1 - A_{v,v}), & \forall u,v \in [n],\ u \neq v \end{cases}$$

**Condition ($\mathcal{C}3$):** Initialization for single node, i.e., node $v$ can reach itself with shortest distance as $0$, and node $v$ is obviously the only node that appears in the shortest path from node $v$ to itself:

$$\begin{cases} r_{v,v} = 1, & \forall v \in [n] \\ d_{v,v} = 0, & \forall v \in [n] \\ \delta_{v,v}^v = 1, & \forall v \in [n] \\ \delta_{v,v}^w = 0, & \forall v,w \in [n],\ v \neq w \end{cases}$$

**Condition ($\mathcal{C}4$):** Initialization for each edge, i.e., if edge $u \to v$ exists, node $u$ can reach node $v$ with shortest distance as $1$. Otherwise, the shortest distance from node $u$ to node $v$ is larger than $1$:

$$A_{u,v} = 1 \Rightarrow r_{u,v} = 1,\ d_{u,v} = 1, \quad \forall u,v \in [n],\ u \neq v$$
$$A_{u,v} = 0 \Rightarrow d_{u,v} > 1, \quad \forall u,v \in [n],\ u \neq v$$

which could be rewritten as the following linear constraints:

$$\begin{cases} r_{u,v} \geq A_{u,v}, & \forall u,v \in [n],\ u \neq v \\ d_{u,v} \geq 2 - A_{u,v}, & \forall u,v \in [n],\ u \neq v \\ d_{u,v} \leq 1 + (n-1) \cdot (1 - A_{u,v}), & \forall u,v \in [n],\ u \neq v \end{cases}$$

**Condition ($\mathcal{C}5$):** Compatibility between distance and reachability, i.e,, node $u$ can reach node $v$ if and only the shortest distance from node $u$ to node $v$ is finite:

$$d_{u,v} < n \Leftrightarrow r_{u,v} = 1,\ \forall u,v \in [n],\ u \neq v$$

which could be rewritten as the following linear constraints:

$$\begin{cases} d_{u,v} \leq n - r_{u,v}, & \forall u,v \in [n],\ u \neq v \\ d_{u,v} \geq n - (n-1) \cdot r_{u,v}, & \forall u,v \in [n],\ u \neq v \end{cases}$$

**Condition ($\mathcal{C}6$):** Compatibility between path and reachability, i.e., (i) if node $w$ appears in the shortest path from node $u$ to node $v$, then node $u$ can reach node $w$, and node $w$ can reach node $v$ (the opposite is not always true), which means that node $u$ can reach node $v$ via node $w$:

$$\delta_{u,v}^w = 1 \Rightarrow r_{u,w} = r_{w,v} = 1 \Rightarrow r_{u,v} = 1, \forall u,v,w \in [n],\ u \neq v \neq w$$

which could be rewritten as the following linear constraints:

$$\begin{cases} r_{u,w} + r_{w,v} \geq 2 \cdot \delta_{u,v}^w, & \forall u,v,w \in [n],\ u \neq v \neq w \\ r_{u,v} \geq r_{u,w} + r_{w,v} - 1, & \forall u,v,w \in [n],\ u \neq v \neq w \end{cases}$$

**Condition ($\mathcal{C}7$):** Construction of shortest path, i.e., (i) always assume that both node $u$ and node $v$ appear in the shortest path from node $u$ to node $v$ for well-definedness, (ii) if edge $u \to v$ exists or node $u$ cannot reach node $v$, then no other nodes can appear in the shortest path from node $u$ to node $v$, (iii) if edge $u \to v$ does not exist but node $u$ can reach node $v$, then at least one node except for node $u$ and node $v$ will appear in the shortest path from node $u$ to node $v$:

$$\delta_{u,v}^u = \delta_{u,v}^v = 1, \qquad \forall u,v \in [n],\ u \neq v$$

$$A_{u,v} = 1 \vee r_{u,v} = 0 \Rightarrow \sum_{w \in [n]} \delta_{u,v}^w = 2, \quad \forall u,v \in [n],\ u \neq v$$

$$A_{u,v} = 0 \wedge r_{u,v} = 1 \Rightarrow \sum_{w \in [n]} \delta_{u,v}^w > 2, \quad \forall u,v \in [n],\ u \neq v$$

Observing that $A_{u,v} = 1 \vee r_{u,v} = 0 \Leftrightarrow r_{u,v} - A_{u,v} = 0$ since $r_{u,v} \geq A_{u,v}$ always holds, we can rewrite these constraints as the following linear constraints:

$$\begin{cases} \delta_{u,v}^u = \delta_{u,v}^v = 1, & \forall u,v \in [n],\ u \neq v \\ \displaystyle\sum_{w \in [n]} \delta_{u,v}^w \geq 2 + r_{u,v} - A_{u,v}, & \forall u,v \in [n],\ u \neq v \\ \displaystyle\sum_{w \in [n]} \delta_{u,v}^w \leq 2 + (n-2) \cdot (r_{u,v} - A_{u,v}), & \forall u,v \in [n],\ u \neq v \end{cases}$$

**Condition ($\mathcal{C}8$):** Triangle inequality of shortest distance, i.e., if node $u$ can reach node $w$ and node $w$ can reach node $v$, then the shortest distance from node $u$ to node $v$ is no larger than the shortest distance from node $u$ to node $w$ then to node $v$, and the equality holds when node $w$ appears in the shortest path from node $u$ to node $v$:

$$\delta_{u,v}^w = 1 \Rightarrow d_{u,v} = d_{u,w} + d_{w,v}, \quad \forall u,v,w \in [n],\ u \neq v \neq w$$

$$r_{u,w} = r_{w,v} = 1 \wedge \delta_{u,v}^w = 0 \Rightarrow d_{u,v} < d_{u,w} + d_{w,v}, \quad \forall u,v,w \in [n],\ u \neq v \neq w$$

where we omit $r_{u,w} = r_{w,v} = 1$ in the first line since $\delta_{u,v}^w = 1$ implies it.

Similarly, we can rewrite these constraints as the following linear constraints:

$$\begin{cases} d_{u,v} \leq d_{u,w} + d_{w,v} - (1 - \delta_{u,v}^w) + (n+1) \cdot (2 - r_{u,w} - r_{w,v}), & \forall u,v,w \in [n],\ u \neq v \neq w \\ d_{u,v} \geq d_{u,w} + d_{w,v} - 2n \cdot (1 - \delta_{u,v}^w), & \forall u,v,w \in [n],\ u \neq v \neq w \end{cases}$$

Putting Conditions ($\mathcal{C}1$)–($\mathcal{C}8$) together presents the final formulation Eq. (Graph-Encoding).

## A.2 THEORETICAL GUARANTEE

All constraints in Eq. (Graph-Encoding) are necessary conditions, i.e., as shown in Lemma 1.

**Lemma 1.** *Given any labeled graph $G$ with $n$ nodes, $\{A_{u,v}(G), r_{u,v}(G), d_{u,v}(G), \delta_{u,v}^w(G)\}_{u,v,w \in [n]}$ is a feasible solution of Eq. (Graph-Encoding) with $n_0 = n$.*

*Proof.* By definition, it is easy to check that $\{A_{u,v}(G), r_{u,v}(G), d_{u,v}(G), \delta_{u,v}^w(G)\}_{u,v,w \in [n]}$ satisfies condition ($\mathcal{C}1$) – ($\mathcal{C}8$). □

The opposite is non-trivial to prove, that is, any feasible solution of Eq. (Graph-Encoding) corresponds to an unique graph with $\sum_{v \in [n]} A_{v,v}$ nodes, which is guaranteed by Theorem 1.

*Proof of Theorem 1.* Denote $\mathcal{F}_{n_0,n}$ as the feasible domain restricted by Eq. (Graph-Encoding) with size $[n_0, n]$, and $\mathcal{G}_{n_0,n}$ as the whole graph space with node numbers in $[n_0, n]$. Define the following mapping:

$$\mathcal{M}_{n_0,n} : \mathcal{F}_{n_0,n} \to \mathcal{G}_{n_0,n}$$
$$\{A_{u,v}, r_{u,v}, d_{u,v}, \delta_{u,v}^w\}_{u,v,w \in [n]} \mapsto \{A_{u,v}\}_{u,v \in [n]}$$

For simplicity, we still use a $n \times n$ adjacency matrix to define a graph with node number less than $n$ and use $A_{v,v}(G)$ to represent the existence of node $v$. Also, the subscriptions, e.g., $\{\}_{u,v,w \in [n]}$, are omitted from now on.

If $n_1 = \sum_{v \in [n]} A_{v,v} < n$, Condition $(\mathcal{C}1)$ forces that:

$$A_{v,v} = \begin{cases} 1, & v \in [n_1] \\ 0, & v \in [n] \backslash [n_0] \end{cases}$$

For any pair of $(u, v)$ with $u \neq v$ and $\max(u, v) \geq n_1$, Conditions $(\mathcal{C}2)$ and $(\mathcal{C}7)$ uniquely define $\{r_{u,v}, d_{u,v}, \delta_{u,v}^w\}$ as:

$$r_{u,v} = 0, \ d_{u,v} = n, \ \delta_{u,v}^w = \begin{cases} 1, & w \in \{u, v\} \\ 0, & w \notin \{u, v\} \end{cases}$$

Therefore, it is equivalent to show that $\mathcal{M}_{n,n}$ is a bijection. Since Lemma 1 already shows that $\mathcal{M}_{n,n}$ is a surjection, it suffices to prove that $\mathcal{M}_{n,n}$ is an injection. Precisely, for any feasible solution $\{A_{u,v}, r_{u,v}, d_{u,v}, \delta_{u,v}^w\}$, there exists a graph $G$ with adjacency matrix given by $\{A_{u,v}(G)\} = \{A_{u,v}\}$, such that:

$$\{r_{u,v}(G), d_{u,v}(G), \delta_{u,v}^w(G)\} = \{r_{u,v}, d_{u,v}, \delta_{u,v}^w\} \quad (\star)$$

Since $r_{v,v}(G), d_{v,v}(G), \delta_{v,v}^w(G), \delta_{u,v}^u(G), \delta_{u,v}^v(G)$ are defined for completeness of our definition, whose variable counterparts are properly and uniquely defined in Condition $(\mathcal{C}3)$ and the first part of Condition $(\mathcal{C}7)$, we only need to consider all triples $(u, v, w)$ with $u \neq v \neq w$, which will not be specified later for simplicity.

Now we are going to prove Eq. $(\star)$ holds by induction on $\min(d_{u,v}(G), d_{u,v}) < n$.

When $\min(d_{u,v}(G), d_{u,v}) = 1$, for any pair of $(u, v)$, we have:

$$\begin{aligned}
d_{u,v}(G) = 1 &\Rightarrow A_{u,v}(G) = 1, \ r_{u,v}(G) = 1, \ \delta_{u,v}^w(G) = 0 && \longleftarrow \text{definition} \\
&\Rightarrow A_{u,v} = 1 && \longleftarrow \text{definition of } \mathcal{M}_{n,n} \\
&\Rightarrow r_{u,v} = 1, \ d_{u,v} = 1, \ \delta_{u,v}^w = 0 && \longleftarrow \text{Conditions } (\mathcal{C}4) + (\mathcal{C}7)
\end{aligned}$$

and:

$$\begin{aligned}
d_{u,v} = 1 &\Rightarrow A_{u,v} = 1, \ r_{u,v} = 1, \ \delta_{u,v}^w = 0 && \longleftarrow \text{Conditions } (\mathcal{C}4) + (\mathcal{C}7) \\
&\Rightarrow A_{u,v}(G) = 1 && \longleftarrow \text{definition of } \mathcal{M}_{n,n} \\
&\Rightarrow r_{u,v}(G) = 1, \ d_{u,v}(G) = 1, \ \delta_{u,v}^w(G) = 0 && \longleftarrow \text{definition}
\end{aligned}$$

For both cases, we have Eq. $(\star)$ holds.

Assume that Eq. $(\star)$ holds for any pair of $(u, v)$ with $\min(d_{u,v}(G), d_{u,v}) \leq sd$ with $sd < n - 1$. Consider the following two cases for $\min(d_{u,v}(G), d_{u,v}) = sd + 1 < n$.

**Case I:** If $d_{u,v}(G) = sd + 1$, we know that $r_{u,v}(G) = 1$ since the shortest distance from node $u$ to node $v$ is finite. For any $w \notin \{u, v\}$ such that $\delta_{u,v}^w(G) = 1$, we have:

$$\begin{aligned}
\delta_{u,v}^w(G) = 1 &\Rightarrow d_{u,w}(G) + d_{w,v}(G) = d_{u,v}(G) && \longleftarrow \text{definition of } \delta_{u,v}^w(G) \\
&\Rightarrow \max(d_{u,w}(G), d_{w,v}(G)) \leq sd && \longleftarrow d_{u,w}(G) > 0, d_{w,v}(G) > 0 \\
&\Rightarrow d_{u,w} = d_{u,w}(G), \ d_{w,v} = d_{w,v}(G) && \longleftarrow \text{assumption of induction} \\
&\Rightarrow r_{u,w} = r_{w,v} = 1 && \longleftarrow \text{Condition } (\mathcal{C}5) \\
&\Rightarrow d_{u,v} \leq d_{u,w} + d_{w,v} = sd + 1 && \longleftarrow \text{Condition } (\mathcal{C}8) \\
&\Rightarrow d_{u,v} = sd + 1 && \longleftarrow d_{u,v} \geq sd + 1 \\
&\Rightarrow r_{u,v} = 1, \ \delta_{u,v}^w = 1 && \longleftarrow \text{Conditions } (\mathcal{C}5) + (\mathcal{C}8)
\end{aligned}$$

which means that $r_{u,v} = r_{u,v}(G)$, $d_{u,v} = d_{u,v}(G)$, $\delta_{u,v}^w = \delta_{u,v}^w(G)$ with $\delta_{u,v}^w(G) = 1$.

For any $w \notin \{u,v\}$ such that $\delta_{u,v}^w(G) = 0$. If $\delta_{u,v}^w = 1$, then we have:

$$\begin{aligned}
\delta_{u,v}^w = 1 &\Rightarrow r_{u,w} = r_{w,v} = 1, \; d_{u,v} = d_{u,w} + d_{w,v} && \longleftarrow \text{Conditions } (\mathcal{C}6) + (\mathcal{C}8) \\
&\Rightarrow \max(d_{u,w}, d_{w,v}) \le sd && \longleftarrow d_{u,w} > 0, \; d_{w,v} > 0 \\
&\Rightarrow d_{u,w}(G) = d_{u,w}, \; d_{w,v}(G) = d_{w,v} && \longleftarrow \text{assumption of induction} \\
&\Rightarrow d_{u,w}(G) + d_{w,v}(G) = sd + 1 = d_{u,v}(G) && \longleftarrow d_{u,v}(G) = d_{u,v} = sd + 1 \\
&\Rightarrow \delta_{u,v}^w(G) = 1 && \longleftarrow \text{definition of } \delta_{u,v}^w(G)
\end{aligned}$$

which contradicts to $\delta_{u,v}^w(G) = 0$. Thus $\delta_{u,v}^w = 0 = \delta_{u,v}^w(G)$ with $\delta_{u,v}^w(G) = 0$.

**Case II:** If $d_{u,v} = sd + 1$, from $sd + 1 > 1$ and Condition $(\mathcal{C}4)$ we know that $A_{u,v} = 0$, from Condition $(\mathcal{C}5)$ we have $r_{u,v} = 1$, and then from Condition $(\mathcal{C}7)$ we obtain that $\sum_{w \in [n]} \delta_{u,v}^w > 2$.

For any $w \notin \{u,v\}$ such that $\delta_{u,v}^w = 1$, we have:

$$\begin{aligned}
\delta_{u,v}^w = 1 &\Rightarrow d_{u,w} + d_{w,v} = d_{u,v} = sd + 1 && \longleftarrow \text{Condition } (\mathcal{C}8) \\
&\Rightarrow \max(d_{u,w}, d_{w,v}) \le sd && \longleftarrow d_{u,w} > 0, d_{w,v} > 0 \\
&\Rightarrow d_{u,w}(G) = d_{u,w}, \; d_{w,v}(G) = d_{w,v} && \longleftarrow \text{assumption of induction} \\
&\Rightarrow d_{u,v}(G) \le d_{u,w}(G) + d_{w,v}(G) = sd + 1 && \longleftarrow \text{definition of } d_{u,v}(G) \\
&\Rightarrow d_{u,v}(G) = sd + 1 && \longleftarrow d_{u,v}(G) \ge sd + 1 \\
&\Rightarrow r_{u,v}(G) = 1, \; \delta_{u,v}^w(G) = 1 && \longleftarrow \text{definition}
\end{aligned}$$

which means that $r_{u,v}(G) = r_{u,v}$, $d_{u,v}(G) = d_{u,v}$, $\delta_{u,v}^w(G) = \delta_{u,v}^w$ with $\delta_{u,v}^w = 1$.

For any $w \notin \{u,v\}$ such that $\delta_{u,v}^w = 0$. If $\delta_{u,v}^w(G) = 1$, then we have:

$$\begin{aligned}
\delta_{u,v}^w(G) = 1 &\Rightarrow d_{u,v}(G) = d_{u,w}(G) + d_{w,v}(G) && \longleftarrow \text{definition of } \delta_{u,v}^w(G) \\
&\Rightarrow \max(d_{u,w}(G), d_{w,v}(G)) \le sd && \longleftarrow d_{u,w}(G) > 0, \; d_{w,v}(G) > 0 \\
&\Rightarrow d_{u,w} = d_{u,w}(G), \; d_{w,v} = d_{w,v}(G) && \longleftarrow \text{assumption of induction} \\
&\Rightarrow d_{u,w} + d_{w,v} = sd + 1 = d_{u,v} && \longleftarrow d_{u,v}(G) = d_{u,v} = sd + 1 \\
&\Rightarrow \delta_{u,v}^w = 1 && \longleftarrow \text{Condition } (\mathcal{C}8)
\end{aligned}$$

which contradicts to $\delta_{u,v}^w = 0$. Thus $\delta_{u,v}^w(G) = 0 = \delta_{u,v}^w$ with $\delta_{u,v}^w = 0$.

The remaining case is $d_{u,v}(G) = d_{u,v} = n$, i.e., node $u$ cannot reach node $v$. It is straightforward to verify that:

$$\begin{aligned}
r_{u,v} = 0 = r_{u,v}(G) && \longleftarrow \text{Condition } (\mathcal{C}5), \text{ definition of } r_{u,v}(G) \\
\delta_{u,v}^w = 0 = \delta_{u,v}^w(G) && \longleftarrow \text{Condition } (\mathcal{C}7), \text{ definition of } \delta_{u,v}^w(G)
\end{aligned}$$

Therefore, Eq. $(\star)$ always holds, which completes the proof. $\qquad\square$

### A.3 Differentiation from prior work

Observe that our encoding Eq. (Graph-Encoding) can be easily restricted to the encoding in Xie et al. (2025) by adding the following constraints:

**Undirected:** Add symmetry constraints to get undirected graphs:

$$A_{u,v} = A_{v,u}, \; r_{u,v} = r_{v,u}, \; d_{u,v} = d_{v,u}, \; \delta_{u,v}^w = \delta_{v,u}^w, \; \forall u, v, w \in [n], \; u < v.$$

**Strong connectivity:** Each existing node can reach all other existing nodes, i.e.,

$$A_{u,u} = A_{v,v} = 1 \Rightarrow r_{u,v} = 1, \; \forall u, v \in [n], \; u \ne v,$$

which can be equivalently rewritten as the following constraint:

$$r_{u,v} \ge A_{u,u} + A_{v,v} - 1, \; \forall u, v \in [n], \; u \ne v.$$

**Remark 1.** *Note that strong connectivity reduces to connectivity for undirected graphs.*

## B KERNEL ENCODING

This section presents encoding for shortest-graph kernels and binary node labels proposed in Xie et al. (2025). Notations are slightly changed to keep consistency with this paper.

**Graph kernel encoding** Introduce indicator variables $p^{u,v}_{s,l_1,l_2} = \mathbf{1}(F_{u,l_1} = 1,\ d_{u,v} = s,\ F_{v,l_2} = 1)$ and count the number of each type of paths as:

$$P_{s,l_1,l_2}(G^i) = \sum_{u,v \in [N]} p^{u,v}_{s,l_1,l_2}.$$

Then SP kernel ($k_g$) could be formulated as:

$$k_g(G, G^i) = \frac{1}{n^2 n_i^2} \sum_{s \in [n], l_1, l_2 \in [L_n]} P_{s,l_1,l_2}(G^i) \cdot P_{s,l_1,l_2},$$

$$k_g(G, G) = \frac{1}{n^4} \sum_{s \in [n], l_1, l_2 \in [L_n]} P^2_{s,l_1,l_2}.$$

To handle the quadratic term $P^2_{s,l_1,l_2}$, we further introduce indicator variables $P^c_{s,l_1,l_2} = \mathbf{1}(P_{s,l_1,l_2} = c)$, and rewrite $k_g(G, G)$ as the following linear form:

$$k_g(G, G) = \frac{1}{n^4} \sum_{s \in [n], l_1, l_2 \in [L_n], c \in [n^2+1]} c^2 \cdot P^c_{s,l_1,l_2}.$$

Before formulating indicators $p^{u,v}_{s,l_1,l_2}$, we need indicators $d^s_{u,v} = \mathbf{1}(d_{u,v} = s)$ that satisfy:

$$\sum_{s \in [n+1]} d^s_{u,v} = 1,\quad \sum_{s \in [n+1]} s \cdot d^s_{u,v} = d_{u,v},\ \forall u, v \in [n],$$

using which we can formulate $p^{u,v}_{s,l_1,l_2}$, $\forall u, v, s \in [n]$, $l_1, l_2 \in [L_n]$ as:

$$3 \cdot p^{u,v}_{s,l_1,l_2} \le F_{u,l_1} + d^s_{u,v} + F_{v,l_2},\ p^{u,v}_{s,l_1,l_2} \ge F_{u,l_1} + d^s_{u,v} + F_{v,l_2} - 2.$$

Similar to $d^s_{u,v}$, indicators $P^c_{s,l_1,l_2}$ can be expressed as:

$$\sum_{c \in [n^2+1]} P^c_{s,l_1,l_2} = 1,\quad \sum_{c \in [n^2+1]} c \cdot P^c_{s,l_1,l_2} = P_{s,l_1,l_2},\ \forall s \in [n],\ l_1, l_2 \in [L_n].$$

**Node label encoding** $k_n$ could be defined in multiple ways, Xie et al. (2025) propose the following permutational-invariant kernel measuring the pair-wise similarity among node features:

$$k_n(F^1_n, F^2_n) := \frac{1}{n_1 n_2 L_n} \sum_{v_1 \in [n_1], v_2 \in [n_2]} F^1_{v_1} \cdot F^2_{v_2} = \frac{1}{n_1 n_2 L_n} \sum_{l \in [L_n]} N_l(F^1_n) \cdot N_1(F^2_n),$$

where $N_l = \sum_{v \in [n]} F_{v,l}$, $\forall l \in [L_n]$, and $n_1 n_2 L_n$ is the normalized coefficient.

Similar to the graph kernel encoding, we have:

$$k_n(F_n, F^i_n) = \frac{1}{n n_i L_n} \sum_{l \in [L_n]} N_l(F^i_n) \cdot N_l,$$

$$k_n(F_n, F_n) = \frac{1}{n^2 L_n} \sum_{l \in [L_n]} N^2_l = \frac{1}{n^2 L_n} \sum_{l \in [L_n], c \in [n+1]} c^2 \cdot N^c_l,$$

where indicators $N^c_l = \mathbf{1}(N_l = c)$ satisfy:

$$\sum_{c \in [n+1]} N^c_l = 1,\quad \sum_{c \in [n+1]} c \cdot N^c_l = N_l,\ \forall l \in [L_n].$$

## C    BENCHMARK-SPECIFIC PARAMETER SETTINGS AND CONSTRAINTS

We further restrict the feasible domain of the proposed encoding in Section 3 to correspond the search space defined by different benchmarks. The following benchmark-specific constraints are added to produce only valid graphs for each benchmark.

**NAS-Bench-101:** Search space for NAS-Bench-101 consists of classic node-labeled DAGs with one source and one sink. NAS-Bench-101 limits the maximal number of edges ($E$) in each cell to 9. After removing the constraint Eq. (1i) on edge labels, the following constraint is added to define the limitation on $E$:

$$\sum_{u<v} A_{u,v} \leq E.$$

NAS-Bench-101 is a particular instance with $n = 7$, $E = 9$, $L_n = 5$, $I = \{0\}$, $O = \{6\}$.

**NAS-Bench-201:** As explained in the main paper, cells in NAS-Bench-201 search space are classic edge-labeled DAGS with one source and one sink. One only need to remove constraints Eqs. (1f)-(1h) for optimization. NAS-Bench-201 is a particular instance with $n = 4$, $L_e = 4$, $I = \{0\}$, $O = \{3\}$. Note that NAS-Bench-201 has 5 labels: one label denotes nonexistence, which is not needed in our encoding.

**NAS-Bench-301:** The search space in NAS-Bench-301 is DARTS which consists of two types of cells: normal cell and reduction cell. Both of them are edge-labeled DAGs with two sources and one sink, but they may not be identical. Although treating them as identical cells in optimization is a common approach (Ru et al., 2021), we treat them together as one disconnected graph to exactly match their original definitions and allow non-identical cells. Denote the normal cell node indices set as $V^n$ and reduction cell node indices set as $V^r$. After removing constraints Eqs. (1f)-(1h), the following constraints are added:

$$d_{u,v} = d_{v,u} = n, \ \forall u \in V^n, \ v \in V^r \tag{2a}$$
$$A_{i,o} = 0, \ \forall i \in I, \ o \in O \tag{2b}$$
$$A_{u,o} = 1, \ \forall u \in V^n \backslash (I \cup O), \ o \in O \cap V^n \tag{2c}$$
$$A_{v,o} = 1, \ \forall v \in V^r \backslash (I \cup O), \ o \in O \cap V^r \tag{2d}$$
$$\sum_{u=0}^{v-1} A_{u,v} = 2, \ \forall v \in [n] \backslash (I \cup O) \tag{2e}$$

Eq. (2a) formulates the disconnected graph structure. Eq. (2b) makes sure no edges between sources and sinks. According to the procedure of formulating cells in DARTS, all the intermediate nodes are connected to the sink within each cell which is formulated as Eqs. (2c)-(2d). Finally, the DARTS search space requires the number of incoming nodes to each intermediate node to be 2, we encode this requirement as Eq. (2e). NAS-Bench-301 is a particular instance with $n = 14$, $L_e = 8$, $I = \{0, 1, 7, 8\}$, $O = \{6, 13\}$, $V^n = \{0, 1, 2, 3, 4, 5, 6\}$, $V^r = \{7, 8, 9, 10, 11, 12, 13\}$. Similar to NAS-Bench-201, NAS-Bench-301 has 8 labels but one label denotes nonexistence, which is not needed in our encoding.

## D    EXPERIMENTAL DETAILS AND FULL RESULTS

### D.1    HYPERPARAMETER SETTINGS IN GP AND BO

We implement our graph kernels defined in Eqs. (linear) and (exponential) as an inherited `Kernel` class in GPflow (Matthews et al., 2017). The initial values of the trainable kernel parameters $\alpha, \beta, \gamma$ and $\sigma_k^2$ are set to 1 with bounds $[0.01, 100]$. In BO, we apply a batch setting to return 5 architectures with the lowest LCB values in each iteration by setting Gurobi parameter `PoolSearchMode`=2. The final MIP model Eq. (Graph-Encoding) is designed for fixed graph size, but NAS-Bench-101 dataset consists of graph sizes ranging from 2 to 7. Our graph encoding supports changeable sizes. The only issue is that the normalized coefficients in kernel encoding are no longer constant, which complicates our formulation. One can resolve this issue by replacing these coefficients by constants or ignoring them. In NAS, however, architectures with more nodes usually have better performance. For instance, most high-quality architectures in NAS-Bench-101 have either 6 or 7 nodes. Therefore, in our experiments for NAS-Bench-101, we simply solve two MIP models with graph size set to $N = 6, 7$ sequentially. Each model returns 5 architectures, we still select 5 of 10 with the lowest

Table 3: NAS algorithms comparison, including whether the method is BO-based, the surrogate model used, and how acquisition function (acq.) is optimized (if BO-based). The superscript '[a]' denotes methods that are not originally designed for NAS but can be adapted for NAS settings. For surrogate models, we use 'v' to denote models using vectorized embeddings of graphs and 'g' to denote models that directly over graph spaces.

| Algorithms | BO-based | Surrogate | Acq. optimization |
|---|---|---|---|
| Random | × | - | - |
| DNGO[a] (Snoek et al., 2015) | ✓ | BNN(v) | mutation |
| BOHAMIANN[a] (Springenberg et al., 2016) | ✓ | BNN(v) | mutation |
| NASBOT (Kandasamy et al., 2018) | ✓ | GP(g) | mutation |
| Evolution (Real et al., 2019) | × | - | - |
| GP-BAYESOPT[a] (Neiswanger et al., 2019) | ✓ | GP(v) | sampling |
| GCN (Wen et al., 2020) | × | - | - |
| BONAS (Shi et al., 2020) | ✓ | GCN(v) | sampling |
| Local search (White et al., 2021b) | × | - | - |
| BANANAS (White et al., 2021a) | ✓ | NN(v) | mutation |
| NAS-BOWL (Ru et al., 2021) | ✓ | GP(g) | mutation |
| NAS-GOAT (ours) | ✓ | GP(g) | MIP |

LCB values. To encourage exploration, we set $\beta_t^{1/2} = 3$ in LCB. The `TimeLimit` parameter in Gurobi for solving each MIP is set as 1800s.

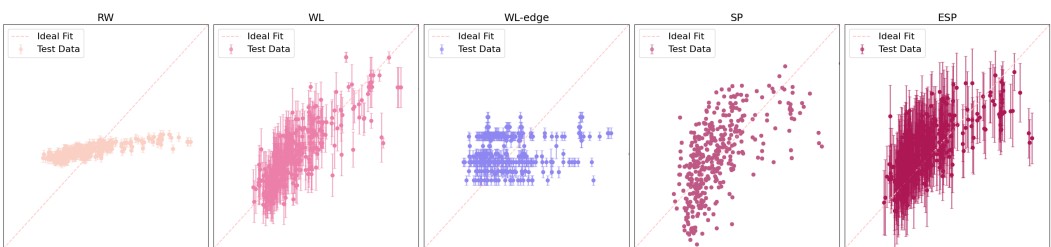

Figure 4: Predictive performance of graph GPs with different kernels. 50 and 400 edge-labeled DAGs are randomly sampled from N201 for training and testing, resp. Predicted deterministic validation error are plotted against the true values, with one standard deviation as error bars.

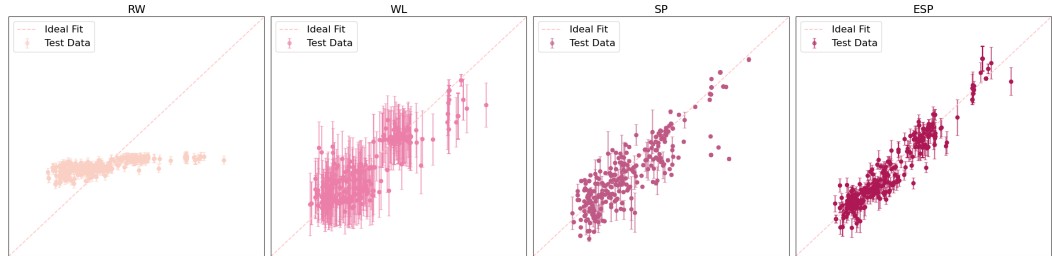

Figure 5: Predictive performance of graph GPs with different kernels. 50 and 400 node-labeled DAGs are randomly sampled from N101 for training and testing resp. Predicted deterministic validation error are plotted against the true values, with one standard deviation as error bars.

## D.2 DETAILS ON BASELINES

We provide more details on algorithms used in Section 4.2. Table 3 summarizes key characteristics of the chosen baselines. We adapt the implementation from White et al. (2020) for all baselines except for NAS-BOWL, where we use the publicly available code from Ru et al. (2021).

- **Random**: Randomly sample the required number of architectures and evaluate them.

- **DNGO**: Deep Network for Global Optimization (DNGO) uses neural networks to learn an adaptive set of basis functions for Bayesian linear regression instead of GP in BO. It is adapted for NAS by treating the adjacency matrix of graph as encoding vector inputs.

- **BOHAMIANN**: Bayesian Optimization with Hamiltonian Monte Carlo Artificial Neural Networks (BOHAMIANN) uses Bayesian neural networks as the surrogate model in both single- and multi-task BO, and achieves scalability through stochastic gradient Hamiltonian Monte Carlo. It is not originally designed for NAS but could be adapted by encoding graph input by adjacency matrix.

- **NASBOT**: Neural Architecture Search with Bayesian Optimisation and Optimal Transport (NASBOT) is a GP-based BO framework for NAS. It defines a distance metric to reveal the similarity between graphs called Optimal Transport Metrics for Architectures of Neural Networks (OTMANN). NASBOT specifically provides a list of operations for the evolutionary algorithm used in the acquisition function optimization.

- **Evolution**: Regularized evolution consists of mutating the best architectures from the population until a given budget runs out. White et al. (2020) set the population size to 30 and outdate the architecture with the worst validation accuracy instead of the oldest one because it results in better performance in NAS tasks following.

- **GP-BAYESOPT**: Standard BO with GP surrogate and UCB acquisition, implemented using ProBO (Neiswanger et al., 2019). Similarity (distance) metric between two architectures is defined as the sum of Hamming distances between the adjacency matrices and the associated operations.

- **GCN**: Use Graph Convolutional Networks (GCN) as the neural predictor to predict the performance of random architectures and select the best $K$ samples for evaluation.

- **BONAS**: Bayesian Optimized Neural Architecture Search (BONAS) uses a GCN as surrogate model in BO to select multiple architectures in each iteration, and apply weight-sharing during the model training to accelerate traditional sampling methods.

- **Local search**: The simplest hill-climbing local search method evaluates all architectures in the neighborhood of a given sample. It is verified by White et al. (2021b) that local search is a strong baseline in NAS when the noise in the benchmark datasets is reduced to a minimum.

- **BANANAS**: Bayesian optimization with neural architectures for NAS (BANANAS) uses a meta neural network over path encoding of individual architectures to predict the validation accuracies. The trained meta NN is used as the surrogate model in BO.

- **NAS-BOWL**: NAS-BOWL is a BO-based NAS algorithm which uses Weisfeiler Lehman (WL) graph kernel in GP surrogate model and adapts to both random sampling and mutation for optimizing the expected improvement (EI) acquisition function. Their experiment results show better performance of NAS-BOWL when using mutation as the acquisition function solver, hence we choose this setting to compare against. NAS-BOWL is considered as the state-of-the-art NAS algorithm.

## D.3 ADDITIONAL GRAPH BO FOR NAS RESULTS

We present additional experiment results on comparing NAS-GOAT with baselines when performing graph BO on NAS-Bench-101, NAS-Bench-201 and NAS-Bench-301 benchmarks. Figure 6 shows the performance of the remaining baselines on CIFAR10 dataset under different benchmarks, where NAS-GOAT shows comparable performance or outperforms others in all cases. Figure 7 shows results on other datasets, where NAS-GOAT continues to show superior performance. NAS-GOAT consistently outperforms other baselines in the most challenging N301 case. We have similar conclusions as in Section 4.4 that NAS-GOAT, as a global graph optimization method, presents more robust performance in terms of the difficulty in the optimization task. Figure 8 summarizes the comparisons between NAS-GOAT and all 11 baselines in terms of test accuracies. NAS-GOAT demonstrates comparable performance as state-of-the-art baselines, e.g. NAS-BOWL, NASBOT, BONAS. Note that in terms of BO performance, only the validation error is the black-box objective, which we expect NAS-GOAT to directly minimize. The test error, on the other hand, is specific to the NAS setting, where a good NAS algorithm is hypothesized to heuristically find architectures

with accompanying good test accuracies, despite potential stochasticity such as overfitting in training process. The performance gap between the two measures could be improved in other benchmarks, e.g., by increasing the size of the validation dataset.

### D.4 COMPARISON AGAINST NON-BO-BASED NAS METHODS

We search for recent NAS algorithms who also include experiments on the N101, N201 and N301 benchmarks and collect their performance metrics in the following tables. We report the average deterministic accuracies. Notice that most of the NAS methods require a larger number of queries to achieve performance comparable to ours, as NAS-GOAT follows a BO framework that maximizes data-efficiency, i.e., returning promising solutions within a limited budget. NAS-GOAT presents comparable performance in N101 and N201 despite the relatively smaller number of queries and outperforms all other methods in the most challenging benchmark N301.

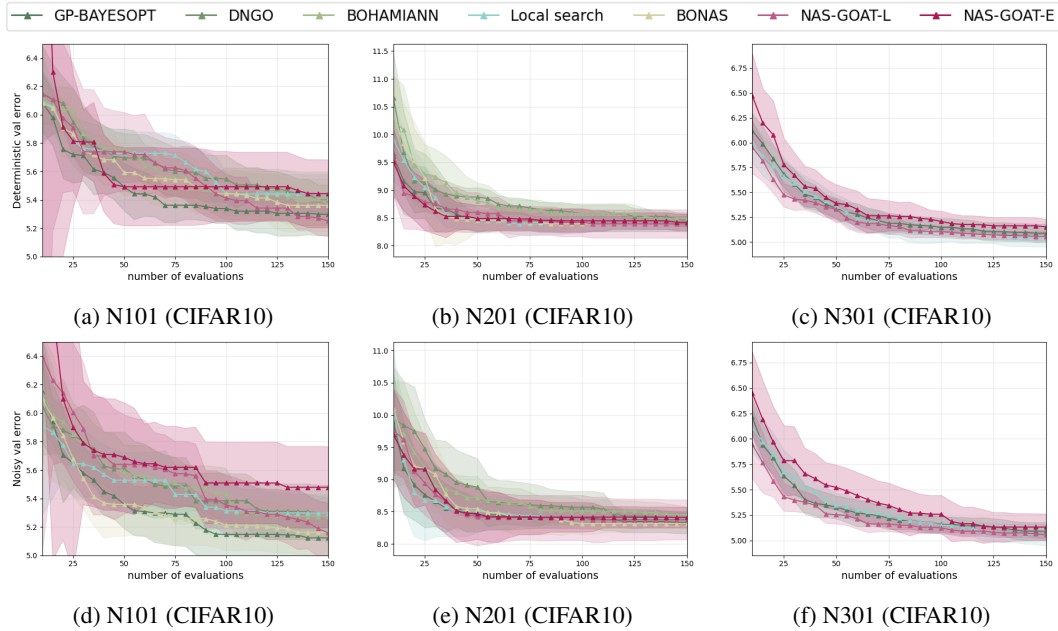

(a) N101 (CIFAR10)  (b) N201 (CIFAR10)  (c) N301 (CIFAR10)

(d) N101 (CIFAR10)  (e) N201 (CIFAR10)  (f) N301 (CIFAR10)

Figure 6: Comparison NAS-GOAT with the remaining baselines. Numerical results of Graph BO on N101, N201 and N301 with CIFAR10 dataset. **Top:** Deterministic validation error. **Bottom:** Noisy validation error. Median with one standard deviation over 20 replications is plotted.

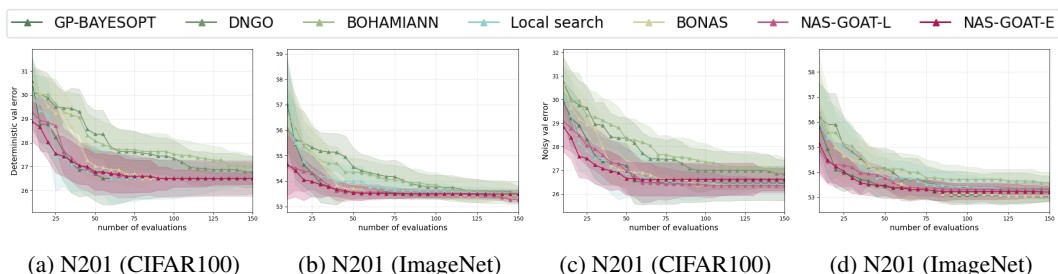

(a) N201 (CIFAR100)  (b) N201 (ImageNet)  (c) N201 (CIFAR100)  (d) N201 (ImageNet)

Figure 7: Comparison NAS-GOAT with the remaining baselines. Numerical results on N201 for other datasets. **(a)-(b):** Deterministic validation error. **(c)-(d):** Noisy validation error. Median with one standard deviation over 20 replications is plotted.

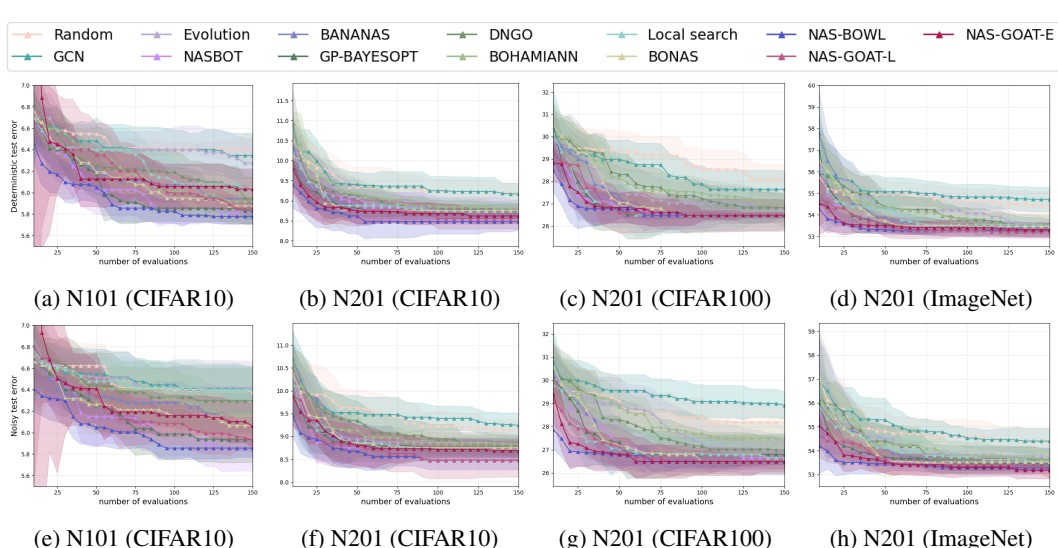

Figure 8: Numerical results on N101 and N201. **Top:** Deterministic test error. **Bottom:** Noisy test error. Median with one standard deviation over 20 replications is plotted.

Table 4: Comparison against non-BO based methods on NAS-Bench-101. Average deterministic validation and test accuracies are reported, with the corresponding sources of the data, number of architectures queried within each algorithm and the number of replications performed.

| Dataset | | CIFAR10 | | | |
|---|---|---|---|---|---|
| Methods | Source | Val | Test | Queries | Replications |
| NAO(Luo et al., 2018) | Asthana et al. (2024) | 94.66 | 93.49 | 192 | 10 |
| SemiNAS(Luo et al., 2020) | Cassimon et al. (2025) | - | 93.89 | 300 | 500 |
| Synflow(Tanaka et al., 2020) | Han et al. (2023) | - | 94.18 | 700 | 5 |
| NASWOT(Mellor et al., 2021) | Cassimon et al. (2025) | - | 91.77 | 100 | 500 |
| WeakNAS(Wu et al., 2021) | Asthana et al. (2024) | - | 94.18 | 200 | 100 |
| GANAS(Rezaei et al., 2021) | Rezaei et al. (2021) | - | 94.23 | 1562 | 10 |
| AG-Net(Lukasik et al., 2022) | Asthana et al. (2024) | 94.90 | 94.18 | 192 | 10 |
| CR-LSO(Rao et al., 2022) | Rao et al. (2022) | - | 94.06 | 500 | 16 |
| CL-fine-tune(Han et al., 2023) | Han et al. (2023) | - | 94.23 | 700 | 5 |
| RAGS-NAS(Xiao & Wang, 2024) | Xiao & Wang (2024) | - | 94.22 | 608 | 10 |
| GraphPNAS(Li et al., 2024) | Cassimon et al. (2025) | - | 94.19 | 300 | 10 |
| DiNAS(Asthana et al., 2024) | Asthana et al. (2024) | 94.98 | 94.27 | 150 | 10 |
| Ape-X(Cassimon et al., 2025) | Cassimon et al. (2025) | - | 93.86 | 150 | 5 |
| NAS-GOAT-L(ours) | - | 94.72 | 94.12 | 150 | 20 |
| NAS-GOAT-E(ours) | - | 94.46 | 93.91 | 150 | 20 |

Table 5: Comparison against non-BO based methods on NAS-Bench-201. Average deterministic validation and test accuracies are reported, with the corresponding sources of the data, number of architectures queried within each algorithm and the number of replications performed.

| Methods | Source | CIFAR10 Val | CIFAR10 Test | CIFAR100 Val | CIFAR100 Test | ImageNet Val | ImageNet Test | Queries | Replications |
|---|---|---|---|---|---|---|---|---|---|
| Synflow(Tanaka et al., 2020) | Han et al. (2023) | - | 94.37 | - | - | - | - | 90 | 5 |
| SGNAS(Huang & Chu, 2021) | Huang & Chu (2021) | 90.18 | 93.53 | 70.28 | 70.31 | 44.65 | 44.98 | - | 3 |
| GANAS(Rezaei et al., 2021) | Rezaei et al. (2021) | - | 94.34 | - | 73.28 | - | 46.80 | 444 | 20 |
| AG-Net(Lukasik et al., 2022) | Asthana et al. (2024) | 91.60 | 94.37 | 73.49 | 73.51 | 46.37 | 46.34 | 192 | 10 |
| CR-LSO(Rao et al., 2022) | Rao et al. (2022) | 91.54 | 94.35 | 73.44 | 73.47 | 46.51 | 46.98 | 500 | 32 |
| $\beta$-DARTS(Ye et al., 2022) | Ye et al. (2022) | 91.55 | 94.36 | 73.49 | 73.51 | 46.37 | 46.34 | - | 4 |
| CL-fine-tune(Han et al., 2023) | Han et al. (2023) | - | 94.37 | - | - | - | - | 90 | 5 |
| RAGS-NAS(Xiao & Wang, 2024) | Xiao & Wang (2024) | 91.61 | 94.37 | 73.51 | 73.49 | 46.64 | 46.61 | 354 | 10 |
| DiNAS(Asthana et al., 2024) | Asthana et al. (2024) | 91.61 | 94.37 | 73.49 | 73.51 | 46.66 | 45.41 | 192 | 10 |
| NAS-GOAT-L(ours) | - | 91.54 | 91.44 | 73.08 | 73.15 | 46.62 | 47.05 | 150 | 20 |
| NAS-GOAT-E(ours) | - | 91.46 | 91.31 | 73.40 | 73.46 | 46.59 | 46.86 | 150 | 20 |

Table 6: Comparison against non-BO based methods on NAS-Bench-301. Average deterministic validation accuracies are reported, with the corresponding sources of the data, number of architectures queried within each algorithm and the number of replications performed.

| Dataset | | CIFAR10 | | |
|---|---|---|---|---|
| Methods | Source | Val | Queries | Replications |
| TPE(Bergstra et al., 2013) | Rao et al. (2022) | 94.50 | 200 | 5 |
| NAO(Luo et al., 2018) | Cassimon et al. (2025) | 94.49 | 200 | 10 |
| Synflow(Tanaka et al., 2020) | Han et al. (2023) | 94.60 | 800 | 5 |
| CMA-ES(Nomura et al., 2021) | Rao et al. (2022) | 94.37 | 200 | 5 |
| AG-Net(Lukasik et al., 2022) | Asthana et al. (2024) | 94.79 | 192 | 10 |
| CR-LSO(Rao et al., 2022) | Rao et al. (2022) | 94.53 | 200 | 5 |
| CL-fine-tune(Han et al., 2023) | Han et al. (2023) | 94.83 | 800 | 5 |
| RAGS-NAS(Xiao & Wang, 2024) | Xiao & Wang (2024) | 94.89 | 300 | 10 |
| DiNAS(Asthana et al., 2024) | Asthana et al. (2024) | 94.92 | 100 | 10 |
| Ape-X(Cassimon et al., 2025) | Cassimon et al. (2025) | 94.83 | 150 | 5 |
| NAS-GOAT-L(ours) | - | 94.94 | 150 | 20 |
| NAS-GOAT-E(ours) | - | 94.85 | 150 | 20 |

