# OpenReview forum: "Global optimization of graph acquisition functions for neural architecture search"
_ICLR.cc/2026/Conference — Submitted to ICLR 2026_

### Official Review · Reviewer_tjo5 · 2025-10-28

**Soundness:** 2
**Presentation:** 3
**Contribution:** 2
**Rating:** 8
**Confidence:** 2

**Summary:**

Briefly summarize the paper and its contributions. You can incorporate Markdown and Latex into your review.
This article proposes an equivalent representation of a general labeled graph in an optimized variable space, where each graph corresponds to a unique feasible solution. It further introduces a universal kernel formula to measure graph similarity, which is compatible with the proposed encoding. This method achieves global acquisition optimization based on graph Bayesian optimization in neural structure search.

**Strengths:**

1.	The paper proposes an equivalent representation of general labeled graphs in the optimization variable space, ensuring that each graph corresponds to a unique feasible solution. Moreover, it introduces a unified kernel formulation that quantifies the similarity between two labeled graphs at the levels of graph structure, node labels, and edge labels.  The advantages over baselines were demonstrated in NAS Bench 101, NAS Bench 201, and NAS Bench 301.
2.	The formulas and derivation proofs in the article are very detailed and accompanied by complete code.

**Weaknesses:**

1.	The benchmarks used (NAS Bench 101, NAS Bench 201, and NAS Bench 301) are all from before 2022. Similarly, the baseline methods such as GCN, NAS BOT, and NAS BOWL are also from before 2021. No experiments were conducted on the latest benchmarks or with more recent baseline methods.
2.	This paper lacks an analysis of the algorithm's time complexity.
3.	The evaluated benchmark is limited to NAS, lacking experiments on real-world tasks, which makes the contribution relatively limited.

**Questions:**

1.	Could experiments be added on more recent and broader benchmarks and baselines?
2.	Could an analysis of the algorithm’s time complexity be provided?

---

> ### Author Response · Authors · 2025-11-21
>
> We appreciate the Reviewer's positive feedback and evaluation of our work. We hope the following clarifications will resolve the Reviewer's concerns and questions:
>
> **[Benchmarks and baselines] (Weakness 1 \& Question 1)**
>
> Please refer to our general response to all Reviewers where we justify our choice of benchmarks. Appendices D.3 and D.4 include comparisons to more recent baselines. For all benchmarks, we included graph BO-based baselines in Figure 6 \& 7 and dozens of more non-BO-based baselines (with publication dates up to 2025) in Tables 4-6. NAS-GOAT shows comparable performance and significantly outperforms others in the most challenging benchmark NAS-Bench-301.
>
> **[Complexity analysis] (Weakness 2 \& Question 2)**
>
> Please refer to our general response to all Reviewers regarding a analysis of the complexity involved in NAS-GOAT.
>
> **[Real-world task] (Weakness 3)**
>
> We are a bit confused by the Reviewer's comment that the ''benchmark is limited to NAS'' as NAS-GOAT serves as a NAS algorithm. We assume the Reviewer is asking about real-world tasks involving training and testing on real data. But the data involved in NAS benchmarks are indeed already generated from actual training/testing on real datasets. Moreover, using known NAS benchmarks enables  reproducibility of our experiments and fairer comparison against other baselines.
> For example, Reviewer uXfa comments that "comprehensive experiments on three major NAS benchmarks under both deterministic and noisy settings demonstrate robustness and efficiency."

---

### Official Review · Reviewer_SBUm · 2025-10-29

**Soundness:** 2
**Presentation:** 3
**Contribution:** 2
**Rating:** 4
**Confidence:** 3

**Summary:**

NAS-GOAT casts cell-based neural architecture search as a Mixed-Integer Program in which graph topology, reachability, shortest-path features and a GP acquisition function are jointly optimized. The resulting MIP is solved to global optimality at every BO step, eliminating hand-crafted mutations and providing certificates of optimality under the surrogate model. Experiments on three public NAS benchmarks demonstrate competitive or superior query efficiency versus recent sampling-based or evolutionary BO baselines.

**Strengths:**

1. The paper is clearly written and easy to follow.
2. The authors design a full condition plan of NAS graph space.
3. The code is supplied, and the hyper-parameters are reported.

**Weaknesses:**

1. The complexity of the method should be analyzed.
2. The main content in Theorem 1 is more likely a modeling plan of the graph space, but it takes too much space in the paper, which makes readers uncomfortable. In addition, Theorem 1 is unnecessary to be a theorem.
3. The experiments are all conducted on NB101~301, it is better to evaluate the method on more datasets. Besides, the method cannot achieve SOTA in some of cases.

**Questions:**

See weakness.

---

> ### Author Response · Authors · 2025-11-21
>
> Thanks for the comments and suggestions. To address the concerns:
>
> **[Complexity analysis] (Weakness 1)**
>
> Please refer to our general response to all Reviewers regarding a analysis of the complexity involved in NAS-GOAT.
>
>
> **[Theorem 1] (Weakness 2)**
>
> Apologies for the confusion: Eq.(Graph-Encoding) is not part of Theorem 1. We will edit the paper structure to make them separately presented in a revised version. We cannot remove the encoding in Eq.(Graph-Encoding), as it is our key contribution. Moreover, Eq.(Graph-Encoding) is not just a ''modeling plan", but rather a complete and equivalent representation of the general graph space without any assumption on the graph structure. Theorem 1 validates this encoding by proving the bijection between the feasible domain restricted by Eq.(Graph-Encoding) and the general graph space. We wish for Theorem 1 to remain a theorem as this bijective relationship between graph space and the feasible space of our MIP is very far from trivial.
>
> **[Benchmarks] (Weakness 3)**
>
> Please refer to our general response to all Reviewers where we justify our choice of benchmarks. We would appreciate if the Reviewer suggested more suitable benchmarks to further validate NAS-GOAT.
>
> **[SOTA] (Weakness 3)**
>
> Tabular benchmarks such as NAS-Bench-101 and NAS-Bench-201 are commonly used to test various NAS methods, and several existing methods can already find near-optimal solutions. The current SOTA BO-based NAS method (to our best knowledge) is NAS-BOWL, which demonstrates good performance and is our most relevant point of comparison. While NAS-GOAT indeed does not exhibit SOTA performance in every single case, our experiments show NAS-GOAT is comparable to or even outperforms NAS-BOWL in many cases, which experimentally confirms: (i) NAS-GOAT can implement global optimization at each iteration, and (ii) global optimality of acquisition optimization results in promising BO results. We present NAS-GOAT as a BO-based NAS method, with main theoretical contributions in terms of the global acquisition optimization as written in the title. Despite our general method and theoretical focus, we already observe SOTA performance in some cases, compared with tailored NAS algorithms (as noted by the Reviewer). We note that in challenging NAS-Bench-301 setting, where the search space is large and contains disconnected graphs, NAS-GOAT significantly outperforms other baselines, highlighting the generality of NAS-GOAT towards challenging search spaces.

---

### Official Review · Reviewer_uXfa · 2025-11-01

**Soundness:** 2
**Presentation:** 3
**Contribution:** 2
**Rating:** 2
**Confidence:** 4

**Summary:**

This paper proposes NAS-GOAT, a framework for globally optimizing graph-based acquisition functions in Bayesian optimization (BO) for neural architecture search (NAS). The authors formulate the graph search space—including reachability, shortest paths, and node/edge labels—as a mixed-integer program (MIP), enabling exact optimization of acquisition functions. The method generalizes prior graph BO formulations (e.g., BoGrape) to handle weakly-connected or disconnected DAGs common in NAS. Experiments on NAS-Bench-101, 201, and 301 show that NAS-GOAT efficiently finds near-optimal architectures, often outperforming or matching state-of-the-art baselines.

**Strengths:**

++ This method extends graph BO to NAS by relaxing the strong connectivity assumption of BoGrape.

++ Comprehensive experiments on three major NAS benchmarks under both deterministic and noisy settings demonstrate robustness and efficiency.

**Weaknesses:**

-- The MIP encoding for graph structures builds heavily on BoGrape, with the main adaptation being the relaxation of strong connectivity. While this is non-trivial, the paper could better highlight what specific constraints were modified or added to handle NAS-specific DAGs.
Specifically, the claim that BoGrape is unsuitable due to strong connectivity is not followed by a clear explanation of how this is resolved beyond "generalizing the graph encoding."

-- I am afraid that this method is not a "plug-and-play" solution. The MIP model must be manually re-derived and re-implemented for each new search space topology. This creates a significant barrier to practical adoption and limits its applicability to new or evolving NAS problems.

**Questions:**

1. I suggest the authors provide more analyze about the differences between this method and BoGrape. As I am concerned, the contribution of this work lies in the adoption of BoGrape for NAS tasks.

---

> ### Author Response · Authors · 2025-11-21
>
> Sincere thanks for noticing the contributions and practical effectiveness of our work. The following responses address the reviewer's concerns:
>
> **[The differences between NAS-GOAT and BoGrape] (Weakness 1)**
>
> Appendix A.3 in current version discusses differences between NAS-GOAT and existing work in BoGrape (Xie et al. 2025). We will further explain here. The MIP encoding in NAS-GOAT is presented in two steps: (i) how to extend existing graph encoding, i.e., BoGrape encoding, to a general setting where the key assumption in BoGrape is removed by adding reachability into the MIP encoding, and (ii) how to restrict the proposed general graph encoding to a NAS-specific graph domain, i.e., DAGs that are weakly-connected (we will explain later why BoGrape encoding cannot handle this setting). Appendix A.1 contains all details of Step (i), and Section 3.2 explains Step (ii). We thank the Reviewer for pointing out that we could better highlight the difference between the NAS-GOAT encoding and the BoGrape encoding, which is not  clear enough in the current version, but crucial to properly highlight our contributions. We will expand the clarification in Appendix A.3 in a revised version. Here we point out that NAS-GOAT is not applying BoGrape to NAS tasks.
>
> **[NAS-GOAT is not an application of BoGrape for NAS tasks] (Question 1)**
>
> NAS-GOAT is a non-trivial generalization of BoGrape rather than an adaption. Note that the NAS-GOAT encoding reduces to the BoGrape encoding by setting all reachability variables $r_{u,v}$ to 1 in Eq.(Graph-Encoding), which effectively means assuming strong connectivity as BoGrape requires. BoGrape assumes strong connectivity since its encoding requires every pair of nodes to be reachable from one another, i.e., $d_{u,v}<n$ for all $u,v$. But, by the definition of a DAG, if u can reach v, then v can never reach u (e.g., directional information flow through a neural network). Therefore, BoGrape is unsuitable for NAS tasks by definition, and cannot be applied in NAS tasks without fundamental changes in the graph encoding. On the other hand, our proposed graph encoding does not have restrictions over the graph structures. Specifically, in the proposed NAS-GOAT encoding, each shortest distance variable $d_{u,v}$ is controlled by its corresponding reachability variable $r_{u,v}$. As shown in Condition ($\mathcal C 5$), if $u$ can reach $v$, then $d_{u,v}$ has a meaningful value, i.e., the shortest distance from $u$ to $v$. Otherwise, $d_{u,v}$ is set to $n$, i.e., $\infty$, indicating that this shortest path does not exist. This generalization makes NAS-GOAT encoding suitable to both weakly connected graphs (like DAGs in NAS-Bench-101 and NAS-Bench-201), and disconnected graphs (like DAGs with two components in NAS-Bench-301).
>
> **[NAS-GOAT is not a "plug-and-play" solution] (Weakness 2)**
>
> To the best of our knowledge, no individual NAS method is a plug-and-play solution. Even for the most general method, random sampling, each sampled architecture must be validated according to different problem settings before being proposed for evaluation. Different NAS tasks have different search space topologies, as well as different features over nodes and/or edges. For a fair comparison of their abilities to explore the search space and find promising solutions, each method must be more-or-less tailored to different tasks. What we can do to decrease the difficulties of practical adoption is to introduce and implement NAS-GOAT for the most general setting from a graph optimization viewpoint: finding a DAG with one or more source(s)/sink(s), node labels, and edge labels. For the sake of exposition, we explicitly describe how to formulate each possible scenario. But for our implementation (the code will be included on publication), users do not need to manually re-derive and re-implement these constraints. Instead, they can simply set suitable hyper-parameters controlling the number of nodes/sources/sinks/node labels/edge labels, and our repository can formulate the search space automatically. In this sense, NAS-GOAT is very close to "plug-and-play".

---

### Author Response · Authors · 2025-11-21
**Response to common comments**

Sincere thanks to all Reviewers for the helpful and detailed comments provided. First, we address the following points that are mentioned by more than one Reviewer:

**[Complexity analysis]**

Complexity analysis can refer to two different NAS-GOAT components: (1) complexity of calculating the kernel values and (2) complexity of solving the acquisition optimization problem (MIP):

(1) The computational complexities of calculating the graph kernel values on a graph with size $n$ are:

- Random Walk (RW): $O(n^3)$

- Weisfeiler-Lehman (WL): $O(hm)$ where $h$ is the number of iterations and $m$ is the number of edges

- Shortest Path (SP): $O(n^3)$

While there are graph kernels with lower complexities than the shortest-path kernel, the complexity in computing kernel values is much less significant than the complexity of the graph optimization (MIP).

(2) For the acquisition optimization, MIP formulations for other graph kernels are not established, making the comparison or complexity analysis over different graph kernels unrealistic.

Solving large MIPs is indeed a long-standing challenge, but modern MIP solvers effectively reduce the solving time by combining many algorithms and best practices. For the scope of NAS-GOAT, cell-based NAS does not involve very large graphs, making global acquisition optimization useful (from an application perspective) and tractable (from a computational perspective). When graph size increases and global optimization becomes computationally intractable, NAS-GOAT is still useful because we can limit the solver runtime and still obtain good feasible solutions with theoretical quality guarantees, i.e., the primal-dual gap. We will add this discussion of complexity into the Appendix of a revised version to help readers better understand our method.

**[Choice of benchmarks]**

We conducted experiments on NAS-Bench-101, NAS-Bench-201 and NAS-Bench-301 as these benchmarks are classic choices in NAS research. Moreover, together they cover the most common and challenging cases in NAS tasks, including node-labeled DAGs, edge-labeled DAGs and disconnected digraphs with multiple sinks and sources. Section 4.1 introduces the graph types involved in each benchmark. Each benchmark also represents graph spaces with different sizes: NAS-Bench-101 and NAS-Bench-201 serve as tabular benchmarks and NAS-Bench-301 includes the challenging open-domain DARTS search space. These benchmarks result in a comprehensive set of NAS cases (as noted by Reviewer uXfa) to study the performance of NAS-GOAT.

---

### Author Response · Authors · 2025-12-02

We sincerely appreciate all the Reviewers for providing helpful feedback to our work, NAS-GOAT. We would like to provide a summary of NAS-GOAT and the Reviewer discussions.

We propose a global graph acquisition function optimization framework in Bayesian optimization for neural architecture search (NAS), namely NAS-GOAT. We encode the general graph space for NAS to include graph properties such as adjacency, reachability, shortest paths and node/edge labels as decision variables and constraints in a mixed-integer program, which allows global optimization of the
acquisition function over the resulting feasible domain. The proposed encoding is theoretically proven to be a bijection of the actual graph domain. Different from prior work, our encoding avoids the requirement on strong connectivity of graphs and allows additional graph types including weakly-connected DAGs, disconnected graphs and the inclusion of node/edge labels and multiple sinks/sources cases. Note these extensions are non-trivial and crucial to NAS. NAS-GOAT is comprehensively evaluated on NAS benchmarks covering node-labeled, edge-labeled and disconnected DAGs cases, often matching or even outperforming tailored state-of-the-art methods.

NAS-GOAT is commonly appreciated by the Reviewers due to its *comprehensive experiments* demonstrating *robustness and efficiency* (by Reviewer uXfa), detailed *formulas and derivation proofs* (by Reviewer tjo5) and clear writing that is *easy to follow* (by Reviewer SBUm). We also provide additional complexity analysis and discussion on choice of benchmarks in our **Response to common comments**, which we believe address the Reviewers’ common questions. In response to individual misunderstandings, we explained why NAS-GOAT is **not an adoption of prior work** in our response to Reviewer uXfa, and clarified our paper structure to Reviewer SBUm on why the Theorem 1 and the Eq. (Graph-Encoding) that the Reviewer thinks *takes too much space* are in-fact our key contributions and foundational components of the paper.

Finally we would like to thank all the Reviewers and the AC for your time commitment and effort. We appreciate the detailed evaluation which helps us improve our presentation and hope our clarifications help your evaluation of NAS-GOAT.

---

### Meta-Review · Area_Chair_46ex · 2026-01-07

**Summary:**

The paper proposes a mixed-integer programming formulation for globally optimizing graph-based acquisition functions in Bayesian optimization for neural architecture search, with theoretical encoding guarantees and empirical evaluation on standard NAS benchmarks.

**Reviewer Concerns:**

Across the reviews, the work is consistently described as clearly written and technically sound, with detailed derivations and reasonably comprehensive experiments, but reviewers also raise concerns about the novelty relative to BoGrape, the limited and somewhat dated benchmark coverage, the lack of a concrete and convincing time-complexity analysis, and the practical burden of tailoring the method to each new search space. While the authors provide clarifications and promise revisions, reviewers note that key concerns are not fully resolved in the current version.

**Reviewer Scores:**

Therefore, based on the reviewers’ assessments and the remaining unresolved issues after the author response, the recommendation is reject.

---

### Decision · Program_Chairs · 2026-01-26

Reject